# Rescue and quality control of sub-daily meteorological data collected at Montevergine Observatory (Southern Apennines), 1884-1963

Vincenzo Capozzi, Yuri Cotroneo, Pasquale Castagno, Carmela De Vivo, Giorgio Budillon

Department of Science and Technology, University of Naples "Parthenope", Centro Direzionale di Napoli – Isola C4, 80143, Italy

*Correspondence to*: Vincenzo Capozzi (vincenzo.capozzi@uniparthenope.it)

**Abstract.** Here we present the rescue of sub-daily meteorological observations collected from 1884 to 1963 at Montevergine Observatory, located on the Italian Southern Apennines. The recovered dataset consists of three daily observations of the following atmospheric variables: dry bulb temperature, wet bulb temperature, water vapour pressure, relative humidity, atmospheric pressure, cloud type, cloud cover, rainfall, snowfall and precipitation type. The data, originally available only as paper-based records, have been digitized following the World Meteorological Organization standard practices. After a cross-check, the digitized data went through three different automatic quality control tests: the gross error test which verifies if the data are within acceptable range limits; the tolerance test that flags if values are above or below monthly climatological limits which are defined in accordance with a probability distribution model specific for each variable; and the temporal coherency test that checks the rate of change flagging unrealistic jumps in consecutive values.

The result of this process is the publication of a new historical dataset that includes, for the first time, digitized and quality-controlled sub-daily meteorological observations collected since the late 19[th] century in the Mediterranean region north of the 37[th] parallel. These data are critical to enhance and complement previously rescued sub-daily historical datasets in central and northern Mediterranean regions, currently limited to the atmospheric pressure observations only. Furthermore, MVOBS dataset can enrich the understanding of high altitude weather and climate variability and contributes to improve the accuracy of reanalysis products prior the 1950s. Data are available on the NOAA's National Centers for Environmental Information (NCEI) public repository and are associated to a DOI: https://doi.org/10.25921/cx3g-rj98 (Capozzi et al., 2019).

## 1 Introduction

Historical meteorological records stretching back to the 19[th] century are crucial for the comprehension of climate variability and its long-term change. These data provide an essential baseline of past climate, which can be useful for many initiatives and efforts focused on monitoring and adapting to climate change and extreme weather events. Thus, the process of past meteorological data retrieval and digitizing, known as "data rescue", is receiving more and more interest within the scientific community, as proved by the high number of projects funded in the last two decades. Examples are the Atmospheric Circulation Reconstructions over the Earth (ACRE, Allan et al., 2011), the MEditerranean DAta REscue (MEDARE) and the Historical

Instrumental Climatological Surface Time Series of the Greater Alpine Region initiatives (HISTALP, Auer et al., 2007; Brunet et al., 2014a,b). These projects, as well as other actions led by local meteorological agencies and research institutions, have two main tasks: (i) enhance the past weather observations asset (in part still unexplored) by discovering, preserving and digitizing paper-based data, and (ii) ensure through quality control procedures that the recovered data are consistent and fully available to the scientific community.

Most of data rescue initiatives are focused on daily and monthly atmospheric observations used for long-term climatological analysis (i.e. daily maximum, minimum and average temperature and daily/monthly total precipitation), but few actions have been established to recover sub-daily data. High-temporal resolution weather data are pivotal to understand the dynamics related to the atmospheric circulation and to the extreme weather events (e.g. Stickler et al., 2014). Thus, sub-daily data constitute a key-input for reanalysis products, both at global and regional scale. A significant effort to rescue historical sub-
daily data has been made by the Twentieth Century Reanalysis (20CR) and the Uncertainties in Ensembles of Regional ReAnalyses (UERRA) projects. The 20CR reanalysis product is the first ensemble of sub-daily global atmospheric conditions (Compo et al., 2011; Cram et al., 2015; Slivinski et al., 2019) spanning from 1851 to 2014 (20CR version 3). This project supported national and regional initiatives focused on the recovery of sub-daily data extending back to the 19th century. The 20CR reanalysis aimed exclusively at the assimilation of atmospheric pressure observations (Slivinski et al., 2019), thus the
resulting observational database is limited to this variable.

On the other hand, UERRA project aim is to develop an ensemble system of European regional reanalyses for the recent decades (1961-2019) through the recovering of sub-daily surface multiparametric weather records (http://uerra.eu/). In the framework of UERRA project, Ashcroft et al. (2018) identified, for the 1957-2010 period, a relevant scarcity of sub-daily data in different European areas, including southern and eastern Mediterranean countries (Morocco, Algeria, Tunisia, Egypt and
Turkey), Central and Eastern Europe (Germany, Czech Republic, Hungary and Balkan Peninsula) and Scandinavia. To fill these gaps, the rescue of 8.8 million observations covering a period from 1877 and 2012 and encompassing a wide range of meteorological variables was carried on.

Despite this relevant effort, the availability of sub-daily meteorological data collected from the late 19th century remains inadequate over most of the European territory. Lack of data is particularly evident in the Mediterranean region, where few
long-term records are available in digital format despite the rich and unevaluable heritage of historical surface meteorological data (Brunet et al., 2014b; Libertino et al., 2018). Fig. 1a provides a clear evidence of the current situation in the Mediterranean area in terms of sub-daily time series digitized and publicly accessible for the period ranging from late 1800 to 1960s. In this figure are shown the 26 historical stations extracted from the International Surface pressure Databank version 4.7 (ISPDv4) and from Ashcroft et al. (2018) datasets, where sub-daily meteorological data are already available in digital format for the
period 1884-1963. The ISPDv4 is an international repository in which converge several national and international historical data collections (Compo et al., 2019). This database incorporates all the observations assimilated by 20CR version 3 (Slivinski et al., 2019) and constitutes the world's largest database of atmospheric pressure data. The majority of the stations (58%) include only atmospheric pressure data (blue dots) and are mainly located in southern Italy, northwestern Italy and Balkan

regions. The remaining 42% of the stations (red dots) include multiparametric meteorological observations recovered by Ashcroft et al. (2018) and are mainly located in southern Mediterranean regions (Algeria, Tunisia and Cyprus).

Our work aims to partially fill the relevant lack in sub-daily data availability prior 1960s, presenting the rescue of the sub-daily meteorological observations collected at Montevergine Observatory (MVOBS) from 1884 to 1963. MVOBS is located in Campania Region (40.936502° N, 14.729150° E) at 1280 m above sea level (asl) near the top of Partenio mountains, which are part of Southern Italy Apennines (Fig. 1b). The rescued sub-daily dataset includes three daily observations of several atmospheric parameters, namely dry bulb temperature, wet bulb temperature, water vapour pressure, relative humidity, atmospheric pressure, cloud type, cloud cover, rainfall, snowfall and precipitation type.

In accordance with the ISPDv4 database, there are only five other historical weather stations in southern Italy extending back several decades prior 1960s that had performed sub-daily multi-parametric observations and that may supply digitized data: Naples Capodimonte (40.88°N, 14.25°E), Foggia Nigri (41.46°N, 15.54°E), Taranto Ferrajolo, (40.47°N, 17.23°E), Palermo (38.10°N, 13.35°E) and Cagliari (39.20°N, 9.15°E). They are all located in coastal or near-coastal areas and only provide atmospheric pressure data with a temporal resolution of one observation per day (rda.ucar.edu/datasets/ds132.2/index.html?sstn=17606&spart=exact#stationViewer, last access: 29 January 2020). The digitized records available for these stations cover the period 1895-1940, except for the Taranto observatory whose time series spans a very limited time interval (1931-1939): for this reason, it has not been included in Fig. 1a.

In light of the above, the sub-daily data rescue activities carried out until now in southern Italy are incomplete. Furthermore, these datasets available in a digital format are only a small part of the larger amount of meteorological information stored in the original paper archives, both in terms of data temporal resolution and number of measured atmospheric parameters.

The sub-daily multiparametric rescued in this study (green dot in fig. 1) is the first dataset dating back to the late 19[th] century, in the Mediterranean region north of the 37[th] parallel, to have been digitized, quality-controlled and made publicly accessible. Therefore, MVOBS data enhance and supplement historical sub-daily datasets currently available in southern Italy area and in the Mediterranean region, broadening the meteorological parameters spectrum and extending the current knowledge on past climate variability to the inland and mountainous sectors.

The sub-daily meteorological records collected at MVOBS have been recovered by the Department of Science and Technology of the University of Naples "Parthenope" in the framework of EPIMETEO project, which aims to characterize the past and present weather conditions in Campania Region.

As already observed by many authors (Diodato, 1992; Capozzi and Budillon, 2013; Capozzi and Budillon, 2017), using daily meteorological parameters data only, MVOBS gives a crucial contribution to study the climate variability.

Distinguish MVOBS features have been synthetized in the following key-points:

a) MVOBS sub-daily multi-variable dataset offers a rare opportunity to investigate climate features of the Central Mediterranean mountain environment prior to 1950s. Mountainous areas are particularly vulnerable to climate change, which has severe impacts on high-altitude ecosystems and habitat (Abeli et al., 2012). In many areas, a solid assessment of mountains

climate variability propaedeutic to the development of reliable future scenarios is difficult due to the scarcity of available data and information;

b) MVOBS is the only meteorological observatory, among those operating in Apennine regions at elevations above 1000 m asl, providing a climatological time series extending back to the late 19th century. According to Brunetti et al. (2006), some high-altitude (> 1000 m asl) time series comparable in length with MVOBS record can be found in Italy only in the Alpine region;

c) MVOBSV dataset, collected near the top of atmospheric boundary layer, allows a proper and objective characterization of air masses, as well as of atmospheric transients that have driven the Central Mediterranean meteorological scenarios from 19th century to early 1960s;

d) Finally, MVOBS time series has been measured in a location whose features have remained unchanged over time, due to the absence of urban settlements. Therefore, these rescued climatic records can be considered devoid of local non-climatic effects related to urbanization, which may cause inhomogeneity in time series (e.g. Jones et al., 1990).

MVOBS can be considered an ideal site for the study of climate variability in a "local-to-global framework". In other words, MVOBS records, collected in a remote high-altitude location, allows investigating the relationship between local and global scale climate changes. Moreover, MVOBS data can shed light on the mutual interactions between large-scale synoptic flow and local orographic features and how these interactions have changed over time due to the variations and anomalies of atmospheric circulation.

Following the standards suggested by World Meteorological Organization (WMO) and the common practices used in climatological data recovery projects, we have structured the rescue of MVOBS dataset into three different steps:

    i.    Identification of metadata (i.e. the information about the history of the station, the instrumentation, the observation practices, the measured atmospheric parameters and the site condition) and data sources;

    ii.    Digitisation of original paper-based data;

    iii.    Quality control and assurance assessment of digitized data through visual inspection and objective statistical methods.

This paper is organized as follows. Section 2 deals with the first two steps of data rescue, providing information about MVOBS history, metadata, availability of meteorological variables and digitising methodology. Section 3 describes the quality control procedures, whereas Section 4 presents some example of use of this dataset. Concluding remarks are drawn in section 5.

## 2 Materials, data and methods

### 2.1 MVOBS history and measurements practises

The first step of the data rescue process consisted of the identification of metadata that can have an impact on the data collection (i.e. location of instruments, changes in practice etc.). We conducted a careful examination of the bibliographic documents

stored in the MVOBS archive located at the Montevergine Abbey. The metadata of MVOBS sub-daily dataset have been

retrieved from two old hand-written diaries, named "*Le Cronache dell'Osservatorio*". Such manuscripts started in 1938 and contains a year-by-year "observatory chronicle" for the period 1881-1946. In order to trace the MVOBS history before 1938, the authors used different sources: postal correspondence, orally transmitted news and a pamphlet entitled "Nel Cinquantenario dell'Osservatorio Meteorologico di Montevergine 1884-1934" (The fiftieth anniversary of Montevergine meteorological observatory 1884-1934), published in 1934 on the observatory fiftieth anniversary. For the period after 1946, metadata have

been retrieved from the meteorological observations registers and from another pamphlet named "Osservatorio meteorologico Santuario di Montevergine" (Montevergine Abbey meteorological observatory) that was published in 1984 to celebrate the MVOBS centenary. During the metadata recovery process particular attention has been given to the assessment of factors that may cause inhomogeneity in the time series, such as instrument relocation and replacement, as well as change in the personnel responsible for the meteorological observations.

According to the old diaries, the idea of establishing a meteorological *specola* in Montevergine was conceived in 1881 by Father Francesco Denza, a meteorologist belonging to the religious order of Barnabiti (i.e. Catholic priests and Religious Brothers belonging to the Roman Catholic religious order of the Clerics Regular of St. Paul). The MVOBS weather observations started three years later on 01 January 1884. MVOBS was part of the first Italian meteorological institution, the "*Italian Central Office of Meteorology and Geodynamics*" (hereafter, Italian Central Office - ICO), established in Rome in

1879 (Maugeri et al., 1998). In the period 1884 - 1894, a room located on the northern front of Montevergine Monastery was dedicated to the weather conditions monitoring. Unfortunately, during this period no specific information about the instruments positioning have been recorded. From 1895, the data were collected in a meteorological tower, built on the eastern side of the Abbey at the suggestion of the ICO. The square-based tower measures 28.4 m in height and 5.7 m in width and was equipped with a Stevenson screen located outside a north-facing window (Fig. 1b). The shelter hosted the following instruments: a

maximum and minimum thermometer, a thermograph, a psychrometer and an evaporimeter. Other instruments, i.e. the rain gauge, the anemograph and the nephoscope were placed on the observatory terrace at the top of the tower. The barometer, the barograph and the recording pluviograph were installed in the observatory room, located at the highest floor of the tower. The nivometric measurements, in compliance with the recommendations of the ICO, were performed through a traditional nivometer consisting of a yardstick and of a snowboard having a section of 0.0001 $m^2$ and an area of 0.01 $m^2$, respectively.

However, no detailed and precise information about the nivometer positioning were found: according to the orally-transmitted information (supplied to the authors of this work by the Benedectine Community of Montevergine), the snow observations may have been collected in the "*Giardinetto dell'Ave Maria*", a cloister of Montevergine Abbey located near the northern side of the meteorological tower. Additional historical photos showing the observatory room, the meteorological tower and the surrounding environment, as well as a modern panoramic view of Montevergine Abbey, are offered in Appendix A.

The weather observations were performed by the Benedictine community of Montevergine, under the guidance of a monk (the Observatory Director) trained by the ICO.

Table 1 provides a summary of the metadata information retrieved for the investigated period (1884-1963). Unfortunately, no relevant and useful information were found for the period 1896 - 1926, except for a change in observatory guidance in 1919. The diaries documented the restoring of the observatory rooms in 1926, and the replacements of some instruments in 1927, 1931, 1937, 1941, 1945 and 1958. From November 1930 to April 1931, the observatory activities were temporarily interrupted. In 1938, MVOBS also became part of the "*Regia Aeronautica*" meteorological network (the modern "*Meteorological Service of the Italian Air Force*"). The observatory served the military institution until March 1952, producing and transmitting six additional bulletins per day (unfortunately not preserved in the observatory archive).

According to the norms prescribed by the ICO, from 1884 to 1932 sub-daily meteorological observations were measured at 09:00, 15:00 and 21:00 hours (local time). From 1933 new standards were adopted by the ICO and the sub-daily data were recorded until 1963 at 08:00, 14:00 and 19:00 hours. These observations include sixteen different meteorological parameters (see Fig. 2 for data availability periods). Near-continuous observations (i.e. measurements characterized by an availability greater than 90%) are available for the following variables: dry bulb temperature, wet bulb temperature, water vapour pressure, relative humidity, atmospheric pressure, cloud type, cloud cover, accumulated rainfall and snowfall and precipitation type. Additional, but incomplete or sporadic, sub-daily observations involved the wind direction and speed, the clouds direction, the snow depth (only recorded in the periods November 1944 - March 1952 and November 1953 - May 1961), the visibility and the low-level clouds base height and quantity (only observed in the period August 1955-May 1961).

In the 1884 - 1963 period, near-continuous daily observations of the following parameters were also carried out: maximum temperature, minimum temperature, accumulated rainfall and snowfall and precipitation event duration. Minimum and maximum temperature were observed at 15:00 and 21:00 (14:00 and 19:00 from 1933), respectively. For short periods, the daily observations also included the evaporation (from 1884 to 1920), the maximum hourly precipitation (from 1941 to May 1961) and the snow depth (from 1937 to 1963).

On May 1964, due to lack of personnel, the meteorological observations for the ICO were suspended. The activities were suppressed until 1968 and then restored in the subsequent year on the proposal of the "*Servizio Idrografico del Genio Civile di Napoli*". However, from 1969 to 2007, only daily observations of the main meteorological variables (maximum and minimum temperature, accumulated rainfall and snowfall and precipitation type) were performed. Since 2008, although the observatory is no more part of any institutional meteorological network, continues its activity through an automatic weather station (AWS) with the scientific support of the Department on Science and Technology of the University of Naples "Parthenope". The AWS has been installed on the observatory terrace and archives several meteorological parameters (temperature, precipitation, pressure, solar radiation, wind direction and wind speed) with a temporal resolution of 1 minute. Nowadays, MVOBS is also equipped with a laser optical disdrometer, mainly used to retrieve measurements of liquid equivalent water snowfall rate (Capozzi et al., 2020).

## 2.2 Data sources and digitisation

The data rescued in this study are stored in eighty paper-based registers, each containing the daily and sub-daily observations collected for each year of the investigated period. A single register consists of 24 pages (two pages for every month) and is formatted in tables according to the standards suggested by the ICO. Fig. 3 presents an extract of the meteorological register for the data measured in March 1892. Each table is related to a month; it is composed of two pages, the first column of each one lists the days, the first row the name of the parameters and observation time. Each box in each column contains the value

of a certain parameter at a specific time of the day. As an example, on the first page (Fig. 3a) the first column (from left) lists the days of the month, whereas the columns from 2 to 10 provide atmospheric pressure observations. For each observation time (09:00, 15:00 and 21:00) are reported the barometer temperature, the measured atmospheric pressure and the corrected atmospheric pressure. The eleventh column is the average of the three-daily corrected pressure observations. Columns from 12 to 17 contain the psychrometric measurements (dry and wet bulb temperature) for each of the three daily observations. The

eighteenth column shows the average of the sub-daily psychrometric records, whereas columns 19 and 20 are dedicated to daily minimum and maximum temperature, respectively. The sub-daily observations of vapour pressure and relative humidity, as well as their daily average, are reported in columns 21-24 and 25-28, respectively. On the following page (Fig. 3b), the first column lists the days of the month. Columns from 2 to 7 show wind direction and speed sub-daily observations. From columns 8 to 10, upper winds (or clouds direction) measurements are listed. Columns from 11 to 19 report sky and weather conditions:

each triplets includes cloud cover, cloud type and hydrometeors observations for a specific time. The hydrometeors column also contains information about the accumulated snowfall and rainfall between two consecutive sub-daily observations. The average of three-daily cloud cover records is listed in the column 20. It is important highlighting that sky conditions, i.e. cloud cover and type, were assessed by visual observations. According to this empirical approach, cloud amount was estimated evaluating the portion of sky covered by clouds. The reference scale for recording the cloud coverage (WMO, 2014) is

expressed in tenths and ranges from 0/10 (sky completely clear) to 10/10 (overcast). To detect the cloud type, various classification methods, which considered cloud species (shape and structure), varieties (cloud arrangement and transparency), as well as cloud level (high, medium and low), were used. Daily accumulated rainfall data and precipitation duration are listed in columns 21-22, whereas daily-accumulated snowfall and evaporation are shown in columns 23-24. Finally, columns 25-26 report ozone observations (never available in MVOBS dataset).

The described data format is representative for the analysed period, except from November 1952 to October 1953 and from March 1962 to December 1963 when a different format was used and the meteorological parameters were sampled only once a day. Moreover, the meteorological registers of 1944-1961 period contain additional columns dedicated to other (sporadically measured) variables, such as snow depth, visibility and low-level clouds base height and quantity. Those registers have a different structure from the standard format described previously. Indeed, a single register consists of 72 pages (i.e. two pages

for every decade of each month) and each page contains two tables. Panels (c) and (d) in Fig. 3 show the register structure for the decade of January 1946. In particular, the upper table on the left page (Fig. 3c) includes three-daily observations of

atmospheric pressure, wind direction and force and cloud direction performed from day 11 to 20. The bottom table shows daily maximum and minimum temperature, three-daily observations of dry and wet bulb temperature, vapour pressure, relative humidity and finally the sum and average of thermometric measurements. The upper table on the right page (Fig. 3d), instead,

contains sub-daily records of the sky conditions (cloud cover and type), accumulated rainfall, snow depth and accumulated snowfall. In addition, daily summaries related to cloud cover, accumulated rainfall and snowfall, maximum 1-hour rainfall amount and precipitation duration (hours and minutes), are reported. The bottom table is dedicated to special notes concerning observed hydrometeors and meteorological phenomena.

The digitisation of the MVOBS dataset available on NOAA's NCEI repository (Capozzi et al., 2019) has been handled by the

personnel involved in this study. A simple "key entry" approach has been used to transcribe the data onto a digital form. Among the techniques generally employed for climate data digitisation, this method is the slower in terms of number of digitized data per hour (Brönnimann et al., 2006); however, at the same time, such method has the lower error rate and follows the standard practises recommended by WMO (2016). The digital templates, developed in Microsoft Excel (Fig. 3e), have been structured in a format that is very similar to the original data source, in order to help the digitizers in keeping track of

their work. Digitized data have been crosschecked with the original source values at the end of every rescued month, to identify and remove transcription errors.

In order to speed-up the digitisation process and to improve the accuracy of the rescued data, we have automatized the transcription of some indirectly measured variables, i.e. the corrected atmospheric pressure, the vapour pressure and the relative humidity. The first one has been determined according to the following relationship (Brombacher et al., 1960):


$$P_0 = P - C_T \qquad (1)$$

Where $P_0$ is the corrected atmospheric pressure (mmHg), i.e. the atmospheric pressure reduced to standard temperature (°C), $P$ is the observed pressure (mmHg) and $C_T$ a temperature correction factor. The latter is defined as:


$$C_T = \frac{-P(a-b)T_b}{1+aT_b} \qquad (2)$$

Where $a$ is the coefficient of expansion for mercury (0.0001818), $b$ the coefficient of linear expansion of brass (0.000184) and $T_b$ is the barometer temperature, measured by an attached thermometer.

Well-known psychrometric formulae have been used to automatically retrieve vapour pressure and relative humidity data. More specifically, the partial pressure of water vapour (*e*) has been obtained as follows (WMO, 2014):

$$e = e_w(T_w) - 6.53 * 10^{-4}(1 + 0.000944 * T_w) * P * (T - T_w) \qquad (3)$$

Where $T$ is the air temperature (dry-bulb temperature), $T_w$ is the wet-bulb temperature and $e_w(T_w)$ is saturation vapour pressure with regard to water at the wet-bulb temperature. $e_w(T_w)$ has been determined according to the following relationship:

$$e_w(T_w) = 6.112 * e^{\frac{(17.62*T_w)}{243.12+T_w}} \qquad (4)$$

Finally, relative humidity ($RH$) has been obtained as $RH = \left(\frac{e}{e_w(T)}\right) * 100$, where $e_w(T)$ is the saturation vapour pressure with regard to water at the dry-bulb temperature.

Moreover, during the digitizing process, special attention was dedicated to the information concerning the equivalent liquid water of the accumulated snowfall. This information, in fact, is essential to reliably characterize the pluviometric regime variability and trend of high-altitude environments (where a large part of winter precipitation falls as snow). The weather
observers operating at MVOBS did not take an unambiguous and homogeneous approach to assess the snow-to-liquid equivalent amount. In addition, in many cases they omitted to note such valuable information. To overcome this issue, in the digital template we created a dedicated column for the Equivalent Liquid Precipitation (ELP). The ELP corresponds to the accumulated rainfall only when the liquid precipitation events occur, whereas in the case of solid precipitation events it represents the amount of liquid precipitation after melting snow. When the observers did not record and note the equivalent in
liquid of snowfall, we manually estimated the ELP using an average snow to liquid water ratio 10:1, which means that the melting of 1 cm of snowfall would produce 1 mm of liquid water (e.g. Winiger, 2005; Egli, 2008; Egli et al., 2009). It is worth bearing in mind that the conversion snow-to-equivalent liquid water is strictly dependent on snowflakes wetness: in wet snow circumstances, the ratio decreases and is on average 5:1, whereas in dry snow conditions the ratio is higher (i.e. it can be 30:1 or greater) because the snow includes a lower liquid water content. Whenever directly measured by observers, the ELP
associated to a determined snowfall event was noted in the original hand-written meteorological registers in the column devoted to accumulated rainfall. To better discriminate between rainfall and equivalent in liquid of solid precipitation, we decided to store such data only in the ELP column of our digital template. The strategy used to assess the ELP parameter is synthetized in Fig. 4, which shows an adapted extract of the rescued MVOBS dataset (available on NOAA's NCEI repository, Capozzi et al., 2019) focusing on the precipitation measurements collected between 28 and 30 January 1956.

**3 Quality control of digitized data**

The third part of our work deals with the development of a rigorous quality control (QC) procedure, aimed to ensure the consistency and the traceability of the rescued sub-daily dataset (WMO, 2008).

Errors and artefacts in historical climatological time series arise mainly from instruments failures, human mistakes during data collection, inaccuracies in manual data transcription on original sources and digitization. As highlighted by Ashcroft et al.

(2018), a comprehensive and reliable QC procedure should be able to identify both systematic and non-systematic errors that may undermine the analysis of climatic signals in a time series.

Steinacker et al. (2011) provided a comprehensive review of the QC strategies usually employed in meteorological field. Depending on the nature of the data, different approaches can be used. As an example, large dataset characterized by high data density in space and time allow the selection of sophisticated QC methods, which are able not only to flag an erroneous value,

but also to correct it through spatial consistency checks. The MVOBS dataset, consisting of observations from a single isolated time series, fits better with simpler QC techniques that can only accept or reject an observation according to objective statistical criteria.

We decided to structure the QC strategy into four different step consisting of a basic visual check, and three different statistical and automatic tests:

i.     The *gross error test*, flags data that are above or below acceptable physical limits. This step also involves an inter-variable check focused on meteorological parameters that are related by physical constraints;

    ii.     The *tolerance test*, detects the outliers, i.e. data that exceed monthly climatological limits defined according to an objective probability distribution model (specific for each of the investigated meteorological parameter);

    iii.     The *temporal coherency test*, identifies unrealistic "jumps" between two consecutive observations, according to the
climatological change that might be expected for a determined variable in a specific time interval.

A schematic diagram of the quality control procedure applied to MVOBS sub-daily data is presented in Fig. 5. According to the results achieved from the different statistical steps, the observation of the parameter collected at a certain time is labelled by a quality flag value from 1 to 3. QC=1 is associated to data that have passed only the gross error test (good data, lower quality level); QC = 2 is the label for records that satisfied both gross and tolerance test (good data, medium quality level) and
finally QC = 3 identifies data that have passed all statistical tests (good data, higher quality level). In summary, data have passed at least one objective statistical check are defined as "good" and are associated to a quality level (ranging from low to high) which is a function of the number of statistical tests passed. Moreover, we flagged as bad data (QC = 8) the records rejected from gross error test and as QC = 9 the measurements identified as suspicious from manual inspection and inter-variable check.

The tolerance and temporal tests require a solid assessment of the climatology of the considered parameter. Therefore, we applied a full QC procedure only to the variables having a high data availability (i.e. at least 30 years of continuous measurements): dry bulb temperature, wet bulb temperature, atmospheric pressure, vapour pressure, relative humidity, cloud cover, rainfall and snowfall. For the remaining parameters (i.e. clouds direction, wind direction, wind speed, cloud type, visibility, low-level cloud base height and quantity, snow depth, precipitation duration and precipitation type), we performed
a basic manual inspection check, that in the case of wind speed, snow depth and visibility takes into account the acceptable limits suggested by WMO (2008).

The applied QC process is exclusively based on the self-consistency and internal coherence of the investigated dataset. In case other sub-daily time series collected in Southern Italy will become available in future, additional quality evaluations of MVOBS sub-daily data will be carried out through spatial consistency checks.

Data homogenization is widely recognized to be an important part of climate data processing (e.g. Alexandersson, 1997). Potential inhomogeneity in MVOBS sub-daily time-series may arise from change of the instruments location occurred in 1895 and from a turnover of the personnel responsible for the meteorological observations (see Table 1). However, the scientific community is still looking for a robust and widely accepted methods for sub-daily data homogenization (Venema et al., 2012). This aspect is well highlighted in the recent work of Ashcroft et al. (2018), in which the rescued sub-daily data were subjected

to a QC procedure that did not include the homogenization.

The following subsections provide details and examples about each of the four QC steps.

### 3.1 Manual inspection

Quality assurance of MVOBS sub-daily dataset starts during the digitisation process. As outlined in Section 2.2, at the end of each rescued month, the values uploaded on digital templates were crosschecked with the ones reported on the original source.

Manual inspection provided a feedback to the digitizers, identifying common typing errors and transcription mistakes (e.g. doubling, adding or forgetting a number, omission of negative sign, forgetting decimal separator etc).

This step also allowed a preliminary assessment of data quality, which helped us to familiarize with some issues affecting the quality of MVOBS sub-daily data, mainly related to the termo-psycrometric observations. Specifically, combining visual checking with plots for data display, we were able to identify residual imprecisions in the digitized measurements that could

have been difficult to identify through the objective statistical procedures.

Through the visual inspection we were able to identify suspicious $T$ and $T_w$ values (flagged with QC = 9) measured from 1920 to 1925 and from 1948 to 1951. In the first period, were only reported even values of the temperature, whereas in the second one were transcribed only the integer part of $T$ and $T_w$.

Other suspicious values identified are associated to observed hydrometeors and the corresponding measurements of

accumulated rainfall or snowfall. In particular, we have examined the coherence between these parameters and we have found two "anomalous" scenarios. In the first scenario of not measurable precipitation, despite the respective accumulated precipitation values is zero, a hydrometeor is detected and reported on meteorological register. Those circumstances were easily recognizable because the observer wrote down the occurrence of no measurable precipitation as textual note. Thus, we did not apply any suspicious or bad quality flag to such data. In the second scenario, although no hydrometeors were detected,

the recorded accumulated precipitation was greater than zero. These measurements accounted for the equivalent in liquid of a snowfall event occurred the day before the specific sub-daily observation was performed. In this case, the accumulated precipitation was measured from the melting of the snow accumulated into the rain gauge and in absence of rain gauge heating this process may take a relatively long time causing an inconsistency between type and accumulated precipitation measurements. To address this issue, we performed a manual correction by aligning in time the occurrence of a certain snowfall

event and the respective ELP value. In some cases, the latter refers to the whole snowfall event (that may span a period that include two or three sub-daily observations and, very occasionally, a period greater than 24-hours), therefore, we were not able to retrieve the sub-daily values.

## 3.2 Statistical tests

As already mentioned, after manual inspection, the digitized sub-daily data went through three different statistical procedures.
The gross error test is the first check and consists in comparing the investigated sub-daily values against their physical limits. This check gives a relevant contribution in quality assurance of climatological dataset, by identifying and discarding clearly erroneous values (Baker, 1992; Feng et al., 2004).

Table 2 lists the upper and lower limits considered for each of the tested meteorological parameters. These limits have been chosen following the WMO (2008) suggestions. These suggestions contain detailed information on the physical boundaries of
several variables. Data passing gross error test have been labelled as QC = 1, otherwise they have been labelled as bad data (QC = 8). This QC step also includes an inter-variable check, aimed to detect and flag inconsistencies between physically related meteorological variables, such as T and $T_w$. In particular, we searched for cases where $T_w > T$ and flagged as suspicious data (QC = 9).

The second statistical procedure is the tolerance test, which aims to detect the outliers defined as extreme values that exceed
climatological limits. The tolerance test has been applied only to sub-daily observations that have passed the manual inspection and the gross error test. In this case, the observation of a certain parameter has been compared to the reference statistical distribution model, computed on monthly basis (i.e. considering all sub-daily observations collected in a specific month during the investigated period, 1884-1963). It should be noted that cloud cover data did not undergo tolerance and temporal coherence tests. The cloud amount was estimated by visual observations using a fixed reference scale. Due to the specific nature of this
parameter and its strong hour-to-hour variability, it is not possible to define climatological limits for outlier and anomalous jumps detection. Therefore, quality control for cloud cover includes only manual inspection and gross error test and it aims to assess the data plausibility and their consistency with other related meteorological parameters, such as cloud type and, when available, low-level clouds base height and quantity.

Depending on the nature of the considered meteorological parameter, we used different statistical distribution model to assess
the climatological limits. Specifically, for variables whose distribution can be fitted by a Gaussian model (i.e. dry and wet bulb temperature, atmospheric pressure and water vapour), an observation $X$ collected at the time $t$ has been flagged as outlier if one of the following inequalities was verified:

$$X_t > 3\sigma + \mu \quad (5.1)$$
$$X_t < 3\sigma - \mu \quad (5.2)$$

Where σ and μ are the reference standard deviation and mean for a specific month. As an example, Figure 6a shows the histogram of the mean January sub-daily dry bulb temperature from 1884 to 1963. This distribution has been easily modelled through the Gaussian probability density function (red curve). Black vertical lines indicate the lower and upper climatological limits (-9.7° C and 10.5° C, respectively) obtained applying the criteria (3σ-μ) and (3σ+μ), with σ = 3.4° C and μ=0.4° C.

In order to represent the sub-daily precipitation data (accumulated rainfall and snowfall) distribution, we employed the gamma function. This function is well suited for positive skewed variables (such as rainfall), as widely recognized in literature (Wilks, 1989; Husak et al., 2007). As already described in section 2.1, the sub-daily meteorological observations performed at MVOBS were not equally spaced over time. Therefore, for time-integrated variables, such as precipitation, an adequate pre-processing has been applied to obtain a fair evaluation of their quality. Specifically, before undergoing the tolerance test, rainfall and, snowfall data have been both divided into two time-series. A first time-series includes the earliest observation of each day collected at 09:00 or 08:00 since 1933, and consists of accumulated precipitation data on a 12 or 13 hours' time interval, respectively. The second time-series encompasses the second and third observations of every day measured at 15:00 and 21:00 from 1884 to 1932 and 14:00 and 19:00 since 1933, reporting the accumulated precipitation on a smaller time interval of 5 or 6 hours. The frequency distribution of the two time-series has been fitted, on a monthly basis, to a gamma distribution after estimating the shape and scale parameters (e.g. Hubbard et al., 2012).

The criterion used to classify a specific precipitation measure $X_t$ as outlier can be written as follows:

$$X_t > T_{PR}(p) \qquad (6)$$

where $T_{PR}$ is the sub-daily precipitation threshold calculated from the Gamma distribution for a given probability $p$ (Hubbard et al., 2012). The latter can be interpreted as the probability that the variable takes a value less than or equal to $T_{PR}$, according to the gamma distribution. In this work, we are interested in flagging the anomalous precipitation events (values that are above the climatological limit), so we considered for both rainfall and snowfall sub-daily data, $p = 0.995$. Figure 6b shows the frequency distribution of the mean November accumulated rainfall (for the first observation of the day) collected from 1884 to 1963. A gamma probability density function has been applied to assess an upper climatological limit (91.2 mm, in this case). Sub-daily values that exceeded such threshold have been classified as outliers.

MVOBS relative humidity data distribution is strongly skewed and, therefore, it is not adequately described by the classic parametric distribution typically employed in meteorological field, such as Gaussian, gamma or generalized pareto. Therefore, we used non-parametric kernel density function to model hygrometric measurements. We applied this estimator to the monthly relative humidity distributions using Gaussian kernel as smoothing function (Wilks, 2006). The threshold for outlier detection has been assessed through the same approach previously described for the precipitation data. However, in this case we are interested to find the lower climatologic limit, so a determined relative humidity observation has been flagged as outlier if:

$$X_t < T_{RH}(p) \qquad (7)$$

where $T_{RH}$ is the threshold on sub-daily relative humidity for a given probability $p$ (=0.005), computed using kernel density distribution. An example of relative humidity frequency distribution modelling is provided by Figure 6c: the latter shows the histogram of May sub-daily relative humidity observations and the associated Kernel probability density function (red curve).

The latter allowed fixing the lower climatological limit (32%), depicted as black vertical line.

To summarize: a sub-daily data that meets (depending on its frequency distribution model) one of the criteria expressed by Eqs. (5.1), (5.2), (6) and (7) has been flagged as outlier and, therefore, its QC value remains at the lowest level (QC = 1); otherwise, is has been labelled as QC = 2.

Sub-daily data that have passed gross and tolerance tests (QC = 2) were subjected to the final QC procedure, the temporal

coherency test. The third statistical test, as the previous two, is considered a fundamental part of any QC procedure (Fiebrich and Crawford, 2001; Steinacker et al., 2011). This check is particularly suitable for high temporal resolution weather data because the correlation degree between time-adjacent samples increases with the sampling rate.

In this work, we applied the temporal coherency test to detect implausible change between instantaneous sub-daily values collected at two consecutive times. Accumulated rainfall and snowfall data, being time-integrated values, were not analysed

in terms of plausible rate of change, so for those parameters the highest quality flag is QC = 2.

The rate of change between two successive measures of a certain parameter has been compared against a maximum climatological gradient ($\Delta$max). The latter has been determined in the following manner: At the first stage, we computed the first derivative of the investigated time series and subsequently we determined on monthly basis the frequency distribution of two different sub-sets: one including only the differences between observations separated by 12 or 13 hours and the other

consisting of differences between data separating in time by 5 or 6 hours. For all the considered meteorological parameters (dry and wet bulb temperature, atmospheric pressure, relative humidity and vapour pressure), the histograms can be modelled using a Gaussian probability density function. Therefore, after retrieving the histogram mean ($\mu_\Delta$) and standard deviation ($\sigma_\Delta$), we fixed $\Delta$max = $\mu_\Delta \pm 3*\sigma_\Delta$: if the rate of change between two consecutive observations (as an example, $\Delta = X_{t+1} - X_t$) lies between such limits, then QC = 3 has been assigned to the record $X_{t+1}$.

Figure 7 shows the sub-daily relative humidity observations collected in March 1901. Each data is color-coded according to the label assigned by QC procedure just described. Rates of change in hygrometric conditions that exceed the maximum climatological gradient assessed for March were detected on March 06 at 09:00 and 15:00 UTC. In the first case, the difference with the previous observation (March 05, 19:00), 49.9%, was greater than $\Delta$max found for 12h observation (39.9%), whereas in the second one the difference with prior record, equal to 42%, is above the climatological limit discovered for 6h data

(35.5%). Both observations were tagged as QC = 2 (red dots). It should be also highlighted that three records go beyond the upper RH physical limit (RH = 100%): such observations were flagged as QC = 8 and are indicated as magenta dots. The remaining data have passed all statistical tests (gross, tolerance and temporal coherency) and they have been labelled as QC = 3 (green dots).

**3.3 Effects of quality control procedure**

Table 3 shows the results of the QC procedure applied to MVOBS sub-daily meteorological data. Among the 5 variables subjected to a full QC, atmospheric pressure is the one with the highest number of values flagged with QC= 3 (98.3%). For this parameter, no bad or suspicious data were recorded. The other four variables (dry and wet bulb temperature, relative humidity and vapour pressure) exhibit similar percentages among different QC levels. Specifically, thermometric measurements have a slightly higher percentage of QC= 3 values, but, at the same time, they show a larger number of

suspicious data after manual inspection (QC = 9). Bad data, i.e. values that exceed acceptable physical limits, have been detected only for two parameters: vapour pressure (0.4%) and relative humidity (0.3%). The amount of data that do not reach the higher QC=3 level is generally less than 2%.

Cloud cover and precipitation went through only the first and second statistical test, respectively. For these parameters, we obtained very good results from the QC analysis. However, for cloud cover we were able to check only the plausibility of data

measured with an empirical approach and for such reason, subject to subtle inhomogeneity and bias caused by changes in personnel involved in the meteorological observations.

Fig. 8a provides a compelling evidence of the QC results on the parameters (i.e. dry and wet bulb temperature, atmospheric pressure vapour pressure and relative humidity) that completed the procedure. The figure shows the distribution over time, computed over a five-year period, of the percentage of sub-daily data whose quality level does not achieve the best value (QC

= 3). The amount of sub-daily values that did not satisfy all steps of QC procedure is generally less than 3%, except in the time segments 1919-1923 (64.8%), 1924-1928 (33.1%), 1944-1948 (17.1%) and 1949-1953 (58.2%). It can be noted that a large part of sub-daily data collected in those periods have been flagged as QC = 9 after visual inspection. The depletion of data quality detected in the above-mentioned sub-periods is mainly caused by human errors already discussed in section 3.1 and, in particular, by imprecisions in thermo-psychrometric measurements. These imprecisions had a negative impact also on the

vapour pressure and relative humidity data quality and, therefore, contribute to a substantial increase in the overall percentage of records flagged as QC = 9. Panels (b) and (c) of Figure 8 show the frequency distribution of dry bulb temperature in 1919-1925 and 1948-1950 period respectively, providing an insight of the inaccuracies in thermos-psychrometric observations found by visual inspection of digitized data. In Fig. 8b, it can be clearly seen that even temperature values have an absolute frequency much higher than odd ones. Whereas, histogram in Fig. 8c, which only shows temperature records between 10 and 20°C,

highlights an anomalous high frequency in the integer temperature values recorded.

This result demonstrates that visual inspection is an essential part of the QC strategy, highlighting some impairments in data quality (mainly caused by human errors) that would otherwise be very difficult to flag through automatic statistical methods.

## 4 Application examples of MVOBS sub-daily dataset

Rescued and quality-controlled historical sub-daily dataset play an invaluable role in many research projects and initiatives focused on the comprehension of climate dynamics and on the identification and analysis of past events severity and frequency (WMO, 2016). However, the traceability of long-term and quality-controlled sub-daily records is currently very poor in Mediterranean area, despite the meteorological conditions have been regularly and thoroughly monitored since 19[th] century (Brunet et al., 2014b). This deficiency exerts a large negative impact not only on studies focused on climate change and

variability, but on initiatives intended to develop socio-economic adaptation strategies. It is a matter of fact that such political actions are based on robustness and accuracy of climate models and reanalysis products, which result in turn from high-quality rescued historical climatic time series.

The recovered data presented in this study offers the first available database, in central and northern Mediterranean region, presenting sub-daily variability of meteorological parameters for a period ranging from late 19[th] century to early 1960s. The

relevance of MVOBS data is also emphasized by the peculiar geographic context in which they have been collected (Apennine region). Therefore, this dataset offers a new opportunity to reconstruct and characterize the past climate and weather event dynamics in high-altitude environments, which are notoriously strongly sensitive to climate change.

In this section, we present some possible uses of the MVOBS sub-daily observations from both a meteorological and climatological point of view. In section 4.1 we focus on the evidence of a past remarkable cold wave event in the MVOBS

data, whereas in section 4.2 we show the potential use of MVOSB data to analyse the long-term atmospheric variability. It is important to notice that, at this stage, sub-daily MVOBS data do not allow to perform a solid climatological analysis, because they have not been homogenized. For this reason, section 4.2 only aims at showing, from a qualitative perspective, some possible future applications of MVOBS sub-daily records in climate fields, with a particular emphasis on some meteorological parameters whose historical variability is largely unknown.


### 4.1 February 1956 cold wave

Fig. 9 shows the behavior of sub-daily meteorological data collected in MVOBS in February 1956. During this period, a strong cold wave affected the Central and Western Europe, as well as many regions of Mediterranean area, causing a large number of fatalities and plant damages (Dizerens et al., 2017). Twardosz et al. (2016) gives an idea of the relevance of such event:

they observe temperature anomalies with respect to the climatological average ranging from -8 to -11°C in most of Europe.

The Fig, 9a presents the evolution of atmospheric pressure (red) and dry bulb temperature (blue) from February 1 to 23. The air temperature was always below 0°C. The cold wave became particularly severe from day 3 to 9 and between day 15 and 18, when the average temperature of -8.4°C and -8.0°C were observed, respectively.

The atmospheric pressure was relatively high, due to the advection of continental arctic air mass. A drop in pressure was

registered between day 10 and 13, when the cold air interacted with moister and warmer Mediterranean air causing large

amount of snow precipitation as shown in the Fig. 9b from the accumulated fresh snow and the snow depth. Snowfall events were recorded on more than half of the total number of sub-daily observations registered in the investigated time interval. Snow depth exhibited a near-continuous positive trend and reached a value just below 300 cm on day 23. Due to the persistence of atmospheric conditions favourable to precipitation events, relative humidity (Fig. 9c, magenta line) was generally higher than 90%. Water vapour pressure values (in black), ranging from 3 to 5 hPa, emphasize the occurrence of conditions closer to saturation.

Fig. 9d supplies a further example of meteorological sub-daily data collected in MVOBS during February 1956, showing useful information about cloud cover (in black) and low-level cloud base height (orange vertical bars).

Fog conditions, which make impossible for a reliable evaluation of cloud type and base altitude, are marked as blue dots.

The qualitative description of these data and their assimilation in large-scale datasets can be useful to the global and regional reanalysis products, which in some cases and in determined regions do not have an adequate input to properly reproduce the dynamics of past meteorological events. In this sense, a relevant example dealing with February 1956 cold wave is provided by the work of Dizerens et al., (2017) that presents the observations assimilated by 20CR reanalysis version 2 (Compo et al., 2011), which covers a period spanning from 1871 to 2010, for 01 February 1956 (12:00 UTC).

In the Central Mediterranean region, data availability on this date is scarce and restricted to the northern Italy and to the coastal areas of Southern Italy and northern Africa. Strong efforts are then needed to fill these consistent gaps: the dataset rescued in our work can certainly give a significant contribution in this direction, providing a unique and rare information on February 1956 cold wave effects on the Apennine Mountains as well as on the thermos-hygrometric conditions near 850 hPa isobaric surface, that is usually used for air masses identification.

## 4.2 Pressure and hail events long-term variability

MVOBS sub-daily data, covering a time interval of 80 years (with some minor gaps), can be helpful for the reconstruction of variability and trend of the first half of the 20th century climate. Specifically, the rescued data can strengthen the current knowledge on the fluctuations over time of both the more studied atmospheric variables, such as air temperature, accumulated rainfall and atmospheric pressure, and the less analysed, i.e. relative humidity, vapour pressure, precipitation type and accumulated snowfall.

Fig. 10 shows the temporal evolution from 1884-1963 of the winter atmospheric pressure (Fig. 10a) and of the yearly hail events frequency occurrence (Fig. 10b) that can help in evaluating the climatological significance of MVOBS sub-daily observations. The first time series has been obtained averaging the sub-daily pressure records collected in the winter season (defined, in this study, as January-February-March) for each year, whereas the second one has been retrieved using the precipitation type information, counting for each year the days in which at least one observation reported a hail event. To highlight from a qualitative perspective, the interannual and decadal variability, a lowess smoothing (red curve) with a span of 10 years has been applied. It should be noted that the displayed time series were not homogenized and, therefore, they may contain break points and artefacts.

Atmospheric pressure data collected at MVOBS are useful to complement and extend the historical sub-daily dataset in Southern Italy, which include observations rescued in four different sites (Fig. 1a). The data from these stations cover the 1895-1940 period and are available, in digital format, with a temporal resolution that is lower than MVOBS dataset; consequently, they totally miss the sub-daily variability of atmospheric parameters. MVOBS can shed more light on past variability of atmospheric pressure in Mediterranean area and can contribute to evaluate its relationship with large-scale atmospheric patterns.

Hail frequency occurrence data offer a great opportunity to build a climatology of hail precipitation, which is gaining more attention due to its severe impacts on crops, properties and buildings (e.g. Santos and Pereira, 2018). Solid and long-term information about hail incidence are available, with few exceptions, only in recent decades (e.g. Zangh et al., 2008; Mezher et al., 2012; Baldi et al., 2014; Santos and Pereira, 2018) and are often subjected to biases due to the different data sources (weather station, insurance companies, newspapers etc.) used for their reconstruction. Data missing is even greater in some countries, such as Italy (Baldi et al., 2014), where the very limited extension of observational network seriously compromises the development of a reliable national hail climatology. In this context, the historical and continuous precipitation type observations performed in MVOBS can supply a relevant contribution for the assessment of past variability of hailfall and of the synoptic and local scale factors that promote the formation of such hydrometeor within convective systems.

## 5. Data availability

MVOBS digitized and quality-controlled dataset is available through NOAA's NCEI historical weather data repository. Two different version of the dataset are provided in Microsoft Excel format. The first one, "Sub_daily_data_MVOBS_raw", includes all the observed parameters without any quality control information, whereas the other one, "Sub_daily_MVOBS_QC_VARIABLES" includes data from all the parameters that have been subjected to statistical quality control tests. Data can be accessed via HTTP using the following website, nodc.noaa.gov/cgi-bin/OAS/prd/accession/download/205785, and are associated to a DOI: https://doi.org/10.25921/cx3g-rj98 (Capozzi et al., 2019).

## 6. Conclusions

Rescued and quality-controlled historical dataset play an invaluable role in many research projects and initiatives focused on the comprehension of climate dynamics and on the identification and analysis of past weather events severity and frequency

(WMO, 2016). The range of applications of this kind of data encompasses many fields and studies, concerning also the socio-economic impacts of climate change, hydrology and agricultural planning.

This manuscript presents the rescue and quality control of sub-daily meteorological observations performed at Montevergine Observatory. The data covers a period spanning from 1884 to 1963 and consists of several variables that provides, three times per day, a complete characterization of the atmosphere state in terms of thermodynamic conditions (dry and wet bulb temperature, atmospheric pressure, vapour pressure and relative humidity), of precipitation type and amount (accumulated snowfall and rainfall) and sky conditions (cloud cover and type). Sub-daily observations and metadata have been recovered

from original hand-written registers preserved in the Montevergine Abbey bibliographic archive, formatted according to the rules of "*Italian Central Office of Meteorology and Geodynamics*", and from old diaries that traces the observatory history. The first step of our work consisted in examining such historic documents to retrieve useful information about the observatory practises, instruments relocation and replacements, change in personnel as well as about data availability. The meteorological records have been digitised using a simple "key-entry" approach, which ensures high-quality standards, despite being the most

time-consuming among the suggested methods by WMO.

Once digitised, sub-daily data have been quality-controlled using a procedure based on the internal consistency and coherence of the dataset, structured into four different stages: manual inspection, gross error test, tolerance test and temporal coherency test. The percentage of observations that satisfy the entire QC chain ranges from 84 to 98%, depending on the considered meteorological variable. Lower data quality (except for atmospheric pressure) have been detected in two time intervals (1920-

1925 and 1948-1951), thanks to the manual inspection that highlighted suspicious data due to human imprecisions. Among the analysed variables, the thermo-psychrometric records proved to be more subject to errors and inconsistencies. This result should not be surprising, considering the many sources of errors that affect the psychrometric measurements (WMO, 2008), mainly related to insufficient ventilation and excessive covering of ice on the wet bulb (an issue that may be particularly common in high-altitude mountainous site).

The scientific community can use the recovered dataset for many purposes, embracing both meteorological and climatological frameworks. This is due to some peculiar features, uncommon for long and old climatological time series, such as the high time-resolution of the weather observations, the variety of recorded meteorological parameters and the uniqueness of the geographical context (southern Apennines Mountains). In the last part of our work, we present two possible uses of MVOBS data, related to the detailed characterization of a severe past weather event (February 1956 cold wave) and to the reconstruction

of variability and trend of winter atmospheric pressure and yearly hail events occurrence frequency.

This work makes available to scientists an old sub-daily climatological dataset for future employments in research activities after its digitation and quality control. In our opinion, more efforts and actions should be designed to recover and valorise old sub-daily records, especially on the Italian territory, which has an inestimable asset of historical meteorological observations. An increase in sub-daily data availability can bring benefits both in terms of quality control and homogenization, allowing

devising procedures relying on spatial consistency and coherence.

In this sense, one of our future aims is to extend the work performed for MVOBS to other ancient observatories of Central and Southern Italy. A primary target of this future research may be the Campania Region in the southern part of Italy. This region has a vivid and rich heritage of past weather data, in large part still unexplored, due to the near-continuous of some meteorological *specola*, including, beside MVOBS, the San Marcellino (Naples) observatory (established in 1860), the

astronomical observatory of Naples Capodimonte (inaugurated on 1821) and the meteorological observatory of Scuola Agraria in Portici (founded in 1898).






**Appendix A: Historical and modern views of Montevergine Observatory**

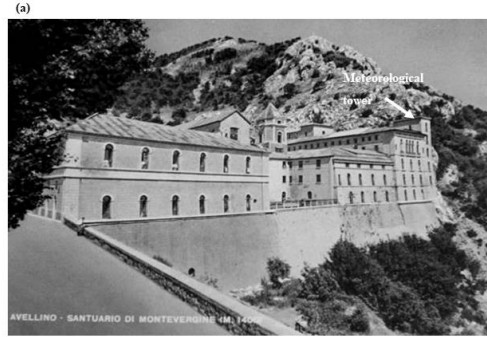 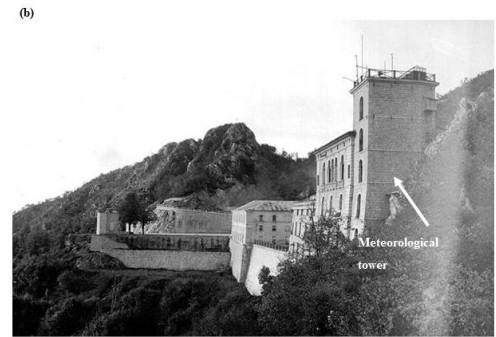

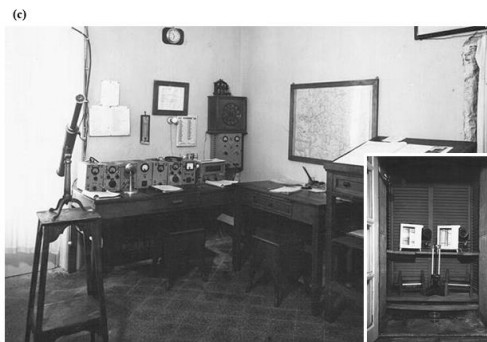 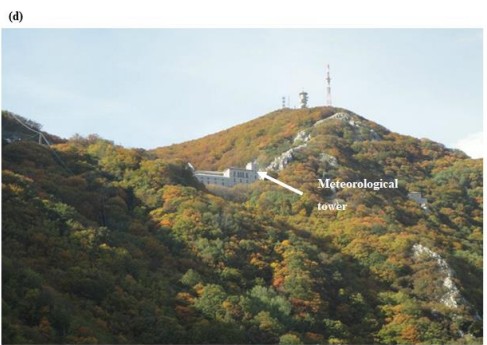

Figure A1: Panel (a) and (b): historical view of MVOBS meteorological tower from a southeastern and a northern direction, respectively. Panel (c): the observatory room in 1950. The small picture in the bottom right corner of the panel shows the inside of Stevenson Screen where thermometric and hygrometric measurements were performed. Panel (d): a recent panoramic view of Montevergine Abbey and MVOBS. Historical and recent images show that MVOBS is surrounded by a natural high-altitude environment, whose features have remained unchanged over time. Photos in panels (a), (b) and (c) are courtesy of the Italian Air Force (www.meteoam.it).

**Author contribution.** VC managed the first two steps of MVOBS data rescue: he analysed the old documents and diaries for metadata retrieval and he digitised the entire sub-daily dataset. CDV analysed MVOBS data availability. YC, PC and VC performed the manual checking of digitized data. YC, VC, PC and CDV designed and applied the statistical test of quality control procedure. VC wrote the manuscript, with contributions of YC, PC, GB and CDV. GB was the coordinator of EPIMETEO project, which provides the funding for this research, and supervised the entire work.

**Competing interests.** The authors declare that they have no conflict of interests.

**Acknowledgments.** The authors of this work are very grateful to the Benedectine Community of Montevergine for affording the opportunity to analyse and digitise the old diaries and meteorological registers stored in Montevergine Abbey. In this respect, we address special thanks to Reverend Father Abbot Riccardo Guariglia and to Father Benedetto Komar, the actual MVOBS Director. Moreover, we are grateful to the Italian Air Force for granting the permission to reproduce, in this manuscript, some old photos of Montevergine Observatory.

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

# TABLES

**Table 1.** Summary of documented metadata in the period from 1884 to 1963. The information about change in instrument relocation and replacement, change in observatory guidance and site measurements conditions were retrieved from the old diaries "*Le Cronache dell'Osservatorio*" and from the meteorological observations registers.

| Time | Metadata information |
|---|---|
| January 1884 | Start of meteorological observations (Observatory Director: Father Giuseppe Llobet) |
| 1888 | Start of meteorological tower building |
| 1893 | End of meteorological tower building |
| 1895 | Meteorological instruments were definitively placed on meteorological tower |
| 1919 | Change in personnel (New Observatory Director: Father Ildebrando Mancini) |
| 1926 | Restoring of observatory rooms |
| 1927 | Change in personnel (New Observatory Director: Father Ilario Mauro) |
| 1927 | Replacement of some meteorological instruments (hair hygrometer, hygrograph, pluviography, solarimeter, thermographs and thermometers) |
| October 1930-April 1931 | Suppression of observatory activities (for undetermined reasons) |
| May 1931 | Change in personnel (New Observatory Director: Father Giulio Corvino) |
| October 1931 | Replacement of maximum thermometer |
| January 1933 | Change in sub-daily observations hours (from 09:00, 15:00 and 21:00 to 08:00, 14:00 and 19:00) |
| 1937 | Change in personnel (New Observatory Director: Father Ugo Inizan) |
| 1937 | Meteorological instrument replacement |
| January 1938 | Change in personnel (New Observatory Director: Father Virginio Cinella) |
| August 1941 | Replacement of Stevenson screen |
| June 1945 | Replacement of pluviograph |
| July 1958 | Replacement of minimum thermometer |

**Table 2.** For each meteorological parameter, upper and lower physical limits used in gross error test are listed.

| Parameter | Upper limit | Lower limit |
|---|---|---|
| Dry bulb temperature | +60°C | -80°C |
| Wet bulb temperature | +60°C | -80°C |
| Atmospheric pressure | 1080 hPa | 500 hPa |
| Vapour pressure | 40 hPa | 0 hPa |
| Relative humidity | 100% | 0% |
| Cloud cover | 10/10 | 0/10 |
| Rainfall | 500 mm | 0 mm |
| Snowfall | 500 cm | 0 cm |

**Table 3.** Results of quality control tests applied to MVOBS sub-daily meteorological data. Each column show the percentage of data flagged as QC = 8, QC =9, QC = 1, QC = 2 and QC = 3. It should be noted that cloud cover data underwent only manual inspection and gross error test, whereas rainfall and snowfall measurements quality was evaluated according to manual inspection, gross error and tolerance tests.

| Parameter | % of QC = 8 bad data | % of QC = 9 suspicious data | % of QC = 1 good data (lower quality level) | % of QC = 2 good data (medium quality level) | % of QC = 3 good data (higher quality level) |
|---|---|---|---|---|---|
| Dry bulb temperature | 0.0 | 13.0 | 0.2 | 0.6 | 86.2 |
| Wet bulb temperature | 0.0 | 13.1 | 0.4 | 0.8 | 85.7 |
| Atmospheric pressure | 0.0 | 0.0 | 0.4 | 1.3 | 98.3 |
| Vapour pressure | 0.4 | 12.8 | 0.4 | 1.1 | 85.3 |
| Relative humidity | 0.3 | 12.8 | 0.4 | 1.6 | 84.8 |
| Cloud cover | 0.0 | 0.0 | 100.0 | Not applied | Not applied |
| Rainfall | 0.0 | 0.0 | 0.1 | 99.9 | Not applied |
| Snowfall | 0.0 | 0.0 | 0.1 | 99.9 | Not applied |

# FIGURES

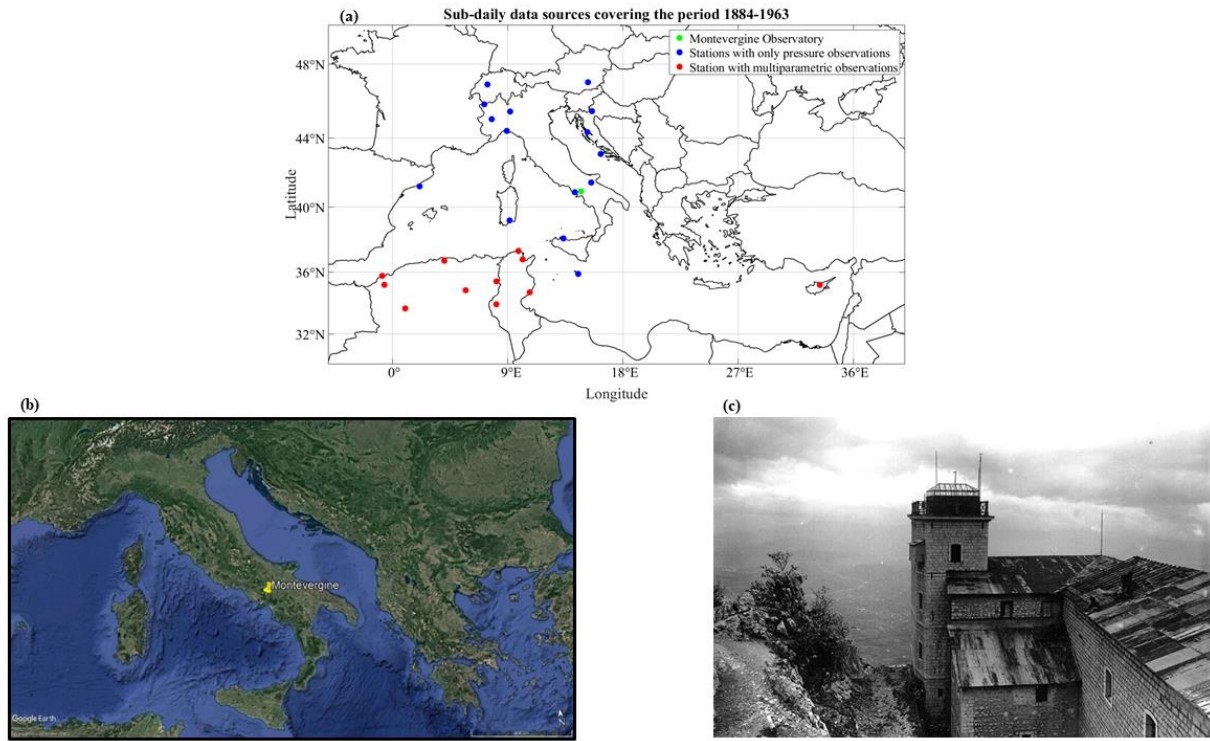

**Figure 1: Panel (a): map of Mediterranean region showing the location of sub-daily meteorological data available in digital format for the period 1884-1963. Blue dots represent the stations including only atmospheric pressure measurements, whereas red dots the one for which multiparametric meteorological observations are available. Data sources have been provided by International Surface pressure Databank version 4.7 (Compo et al., 2019; rda.ucar.edu/datasets/ds132.2/index.html?sstn=17606&spart=exact#stationViewer) and by Ashcroft et al. (2018) datasets. Panel (b) shows Central Mediterranean region, including Montevergine location (highlighted as yellow marker). Montevergine position is also marked on panel (as) as green dot. Image credits: © Google Earth, Data Sio, NOAA, U.S. Navy, NGA, GEBCO. Panel (c) presents an old photo of Montevergine Observatory tower, situated near the top of Partenio mountain chain on the north-eastern side of Montevergine Abbey. Image courtesy of Italian Air Force (http://www.meteoam.it/page/montevergine).**

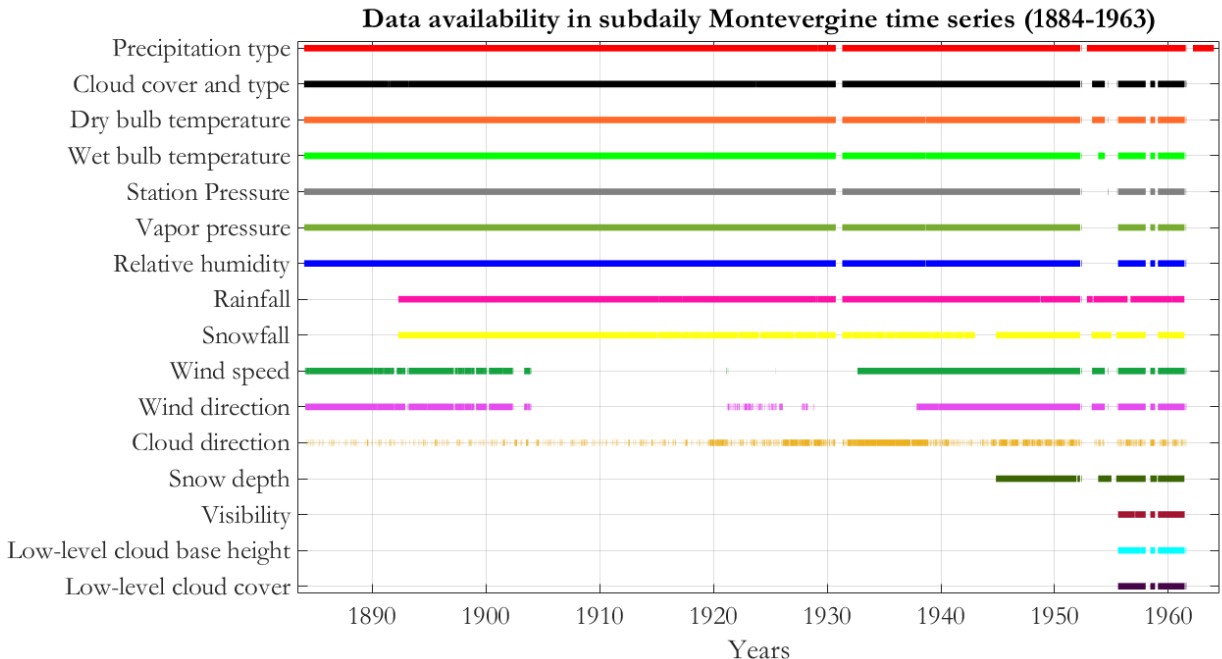

**Figure 2: Data availability of the MVOBS sub-daily dataset in the period ranging from 1884 to 1963. Near-continuous observations are available only for the following variables: precipitation type, cloud cover and type, dry bulb temperature, atmospheric pressure, vapour pressure, relative humidity, rainfall and snowfall.**

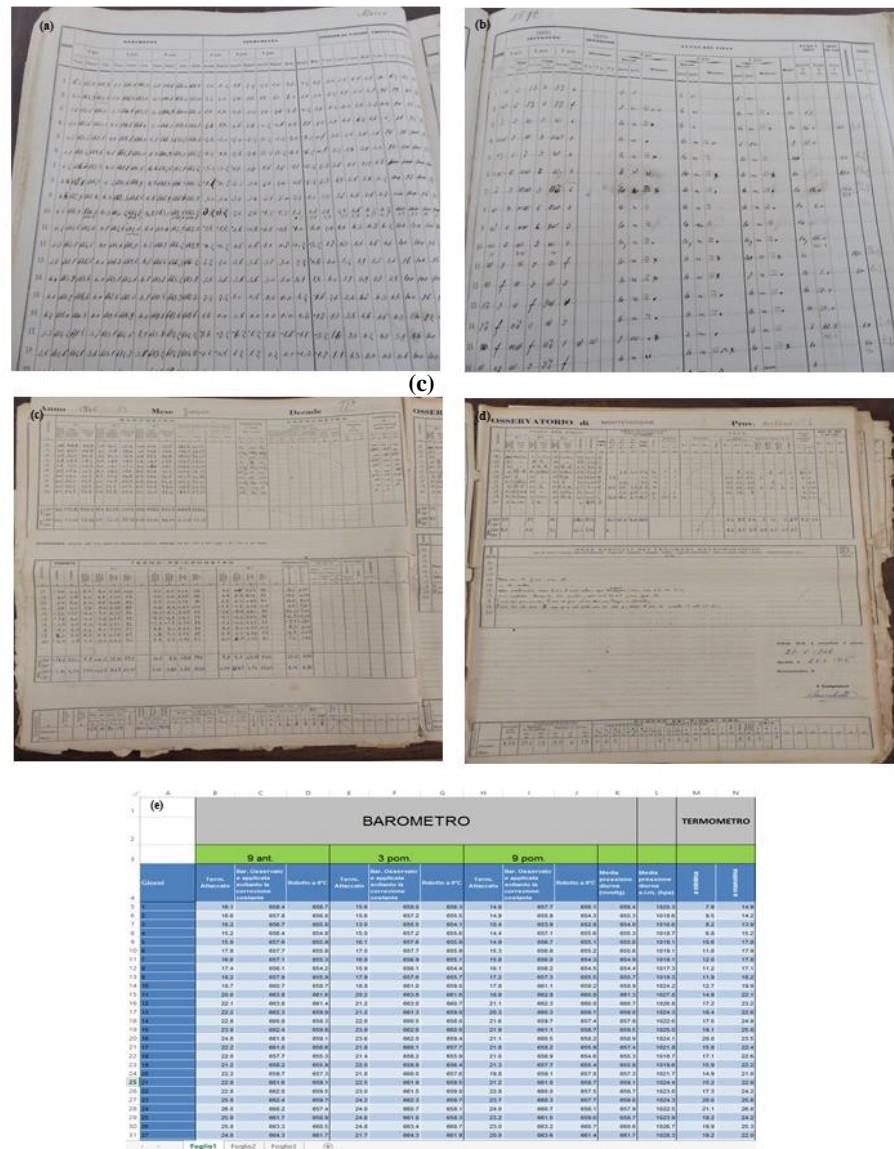


**Figure 3: Upper (a, b) and middle (c, d) panels show an example of original data source (March 1892 and January 1946, respectively). Each row accounts for the observations of a specific day, including their average on decadal and monthly basis, whereas each column is devoted to the records of a determined parameter at a specific hour of the day. The bottom panel (e) is an example (referred to data collected in March 1892) of the template used in the data digitisation. The rows are designed to match the location of the data**
**in the original source.**

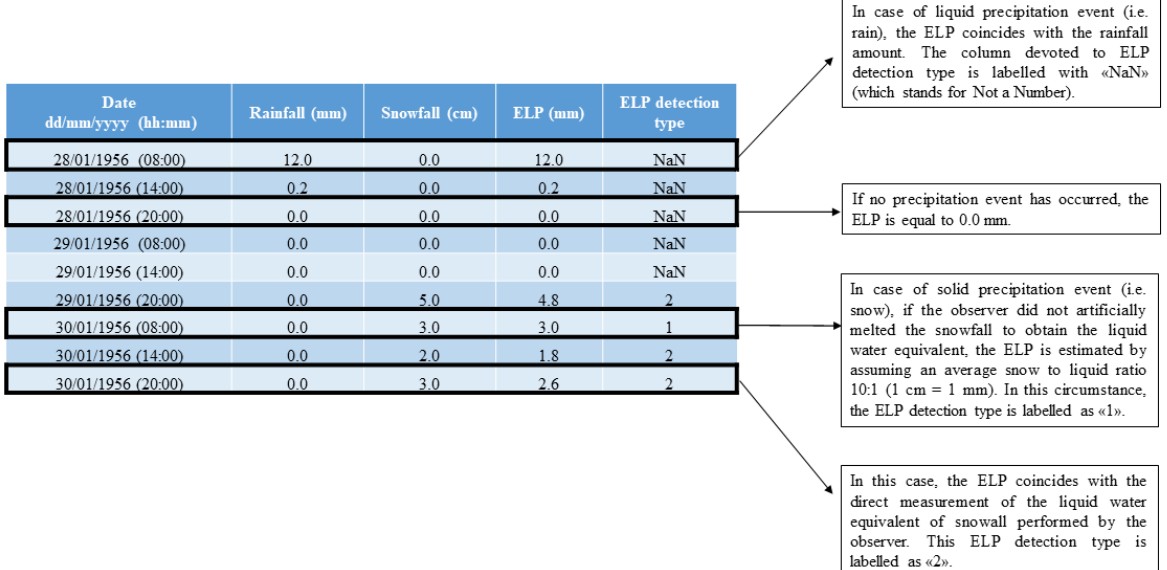

| Date dd/mm/yyyy (hh:mm) | Rainfall (mm) | Snowfall (cm) | ELP (mm) | ELP detection type |
|---|---|---|---|---|
| 28/01/1956 (08:00) | 12.0 | 0.0 | 12.0 | NaN |
| 28/01/1956 (14:00) | 0.2 | 0.0 | 0.2 | NaN |
| 28/01/1956 (20:00) | 0.0 | 0.0 | 0.0 | NaN |
| 29/01/1956 (08:00) | 0.0 | 0.0 | 0.0 | NaN |
| 29/01/1956 (14:00) | 0.0 | 0.0 | 0.0 | NaN |
| 29/01/1956 (20:00) | 0.0 | 5.0 | 4.8 | 2 |
| 30/01/1956 (08:00) | 0.0 | 3.0 | 3.0 | 1 |
| 30/01/1956 (14:00) | 0.0 | 2.0 | 1.8 | 2 |
| 30/01/1956 (20:00) | 0.0 | 3.0 | 2.6 | 2 |

In case of liquid precipitation event (i.e. rain), the ELP coincides with the rainfall amount. The column devoted to ELP detection type is labelled with «NaN» (which stands for Not a Number).

If no precipitation event has occurred, the ELP is equal to 0.0 mm.

In case of solid precipitation event (i.e. snow), if the observer did not artificially melted the snowfall to obtain the liquid water equivalent, the ELP is estimated by assuming an average snow to liquid ratio 10:1 (1 cm = 1 mm). In this circumstance, the ELP detection type is labelled as «1».

In this case, the ELP coincides with the direct measurement of the liquid water equivalent of snowall performed by the observer. This ELP detection type is labelled as «2».

**Figure 4: Summary of the strategy used to assess the Equivalent Liquid Precipitation (ELP) parameter. The left table has been obtained by adapting an extract of the digital version of MVOBS dataset available on NOAA's NCEI repository (Capozzi et al., 2019). It lists the sub-daily precipitation data observed between 28 and 30 January 1956. From left to right, rainfall (mm), snowfall (cm), ELP (mm) and ELP detection type (expressed as numeric or textual label). The rows highlighted in black present different ELP estimation scenarios.**




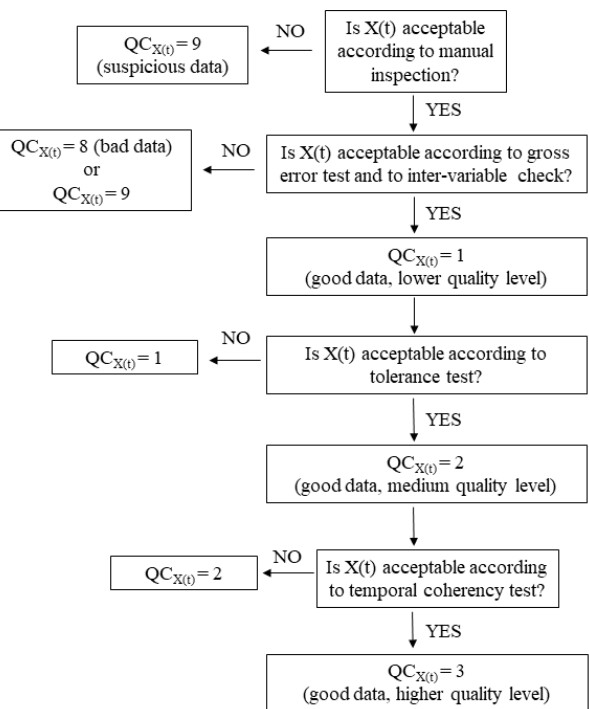

Figure 5: A schematic of the QC strategy developed in this study to check the observation of a determined parameter X collected at the time *t*. It should be highlighted that the cloud cover parameter underwent only gross error test and the temporal coherency test has not been applied to precipitation data (accumulated rainfall and snowfall).

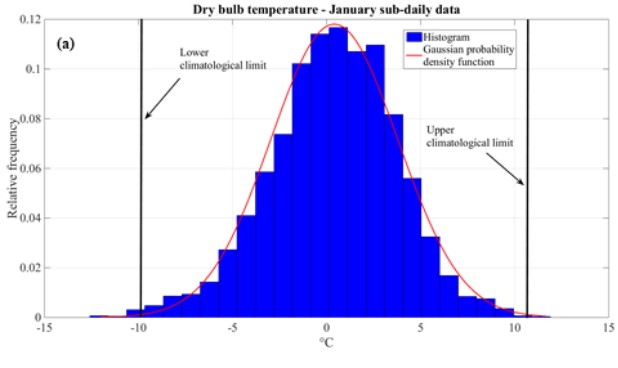

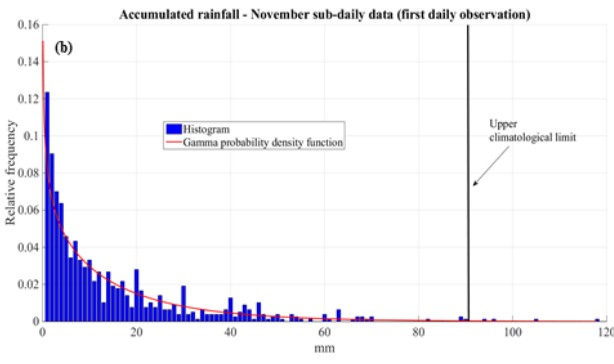

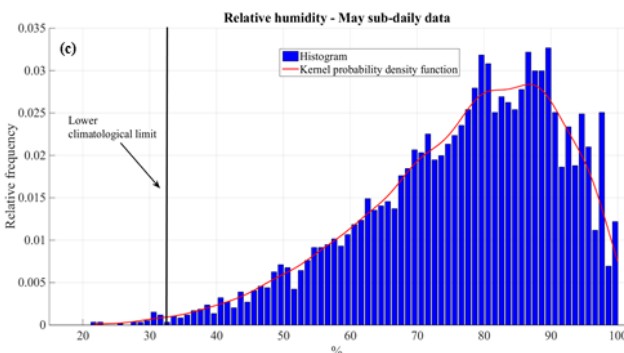

**Figure 6: Examples of probability distribution models designed in the framework of tolerance test. Panel (a) shows the Gaussian probability density function applied to dry-bulb temperature collected in January, panel (b) the gamma distribution density function used to model accumulated rainfall data measured in November and panel (c) the kernel density function applied to relative humidity records collected in May. Black vertical lines indicate the tolerance limits fixed using the criteria expressed from equations (5.1), (5.2), (6) and (7).**


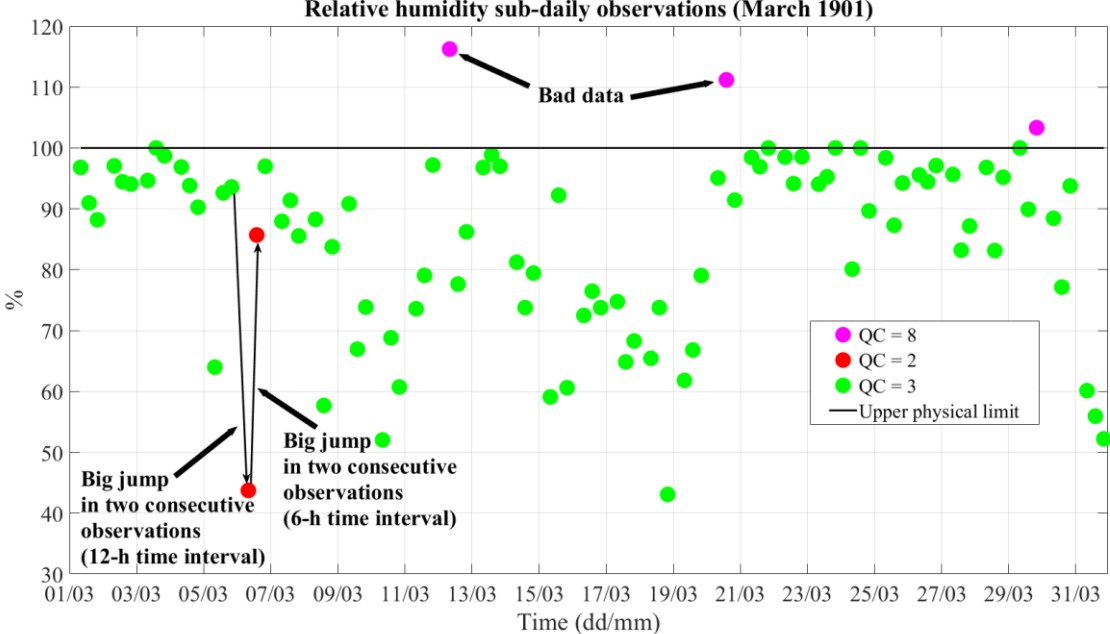

**Figure 7: Relative humidity (in %) sub-daily observations collected in March 1901. Each record is color-coded according to its quality flag: bad data (QC=8; magenta), good data with medium quality level (QC=2; red) and good data with higher quality level (QC=3; green). In this example, no good data with lower quality level (QC=1) and suspicious data (QC=9) were detected. Black horizontal line shows upper physical limit.**

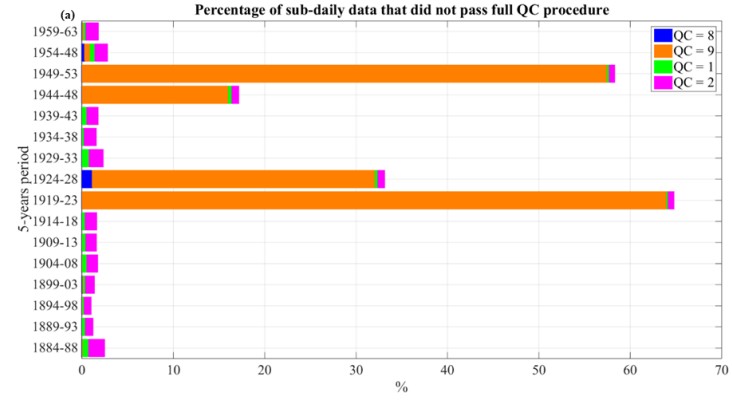

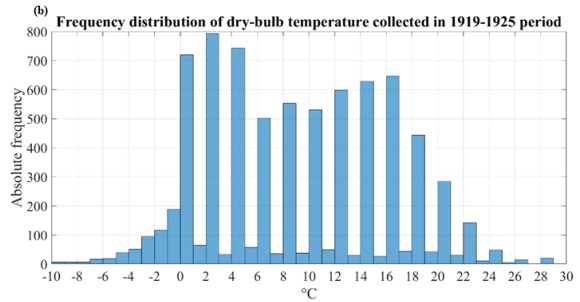

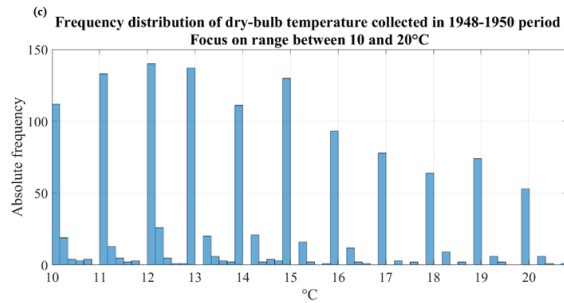

**Figure 8: In panel (a), colour coded bars indicate the distribution over time, computed on five-year period, of the percentage of sub-daily data that did not pass the full QC procedure: QC = 8 (blue), QC = 9 (orange), QC = 1 (green) and QC = 2 (magenta). Panel (b) and (c) present the frequency distribution of dry bulb temperature in 1919-1925 and in 1948-1950 period, respectively. It should be noticed that panel (c) only shows temperature values between 10 and 20° C. The bin width is 1.0°C in panel (b), 0.2°C in panel (c).**

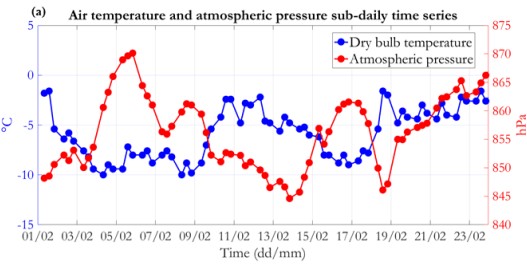
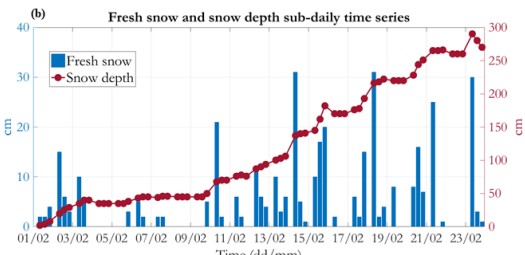

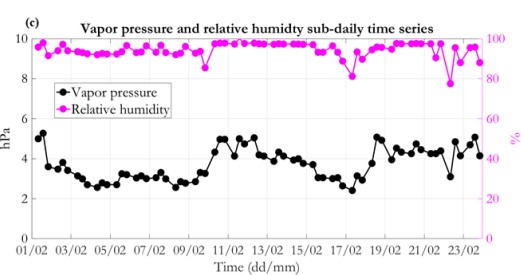
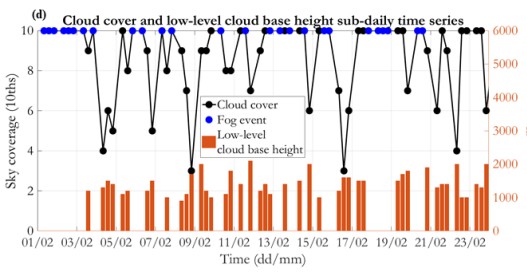


**Figure 9: Sub-daily time series of different meteorological parameters observed from February 01 to 23, 1956. Panel (a) shows the dry bulb temperature and atmospheric pressure. Panel (b) presents the fresh snow and snow depth records. Panel (c) shows vapour pressure and relative humidity. Panel (d) plots the cloud cover and low-level cloud base height observations. It should be noticed that such data were subject to a quality control procedure that did not include the homogenization.**


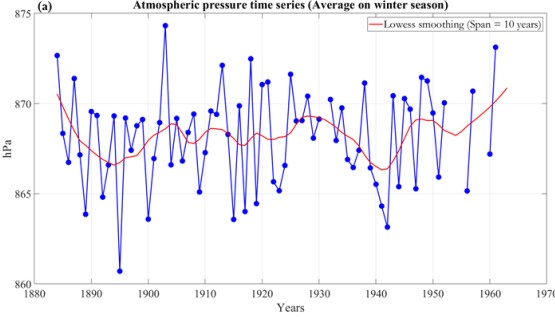

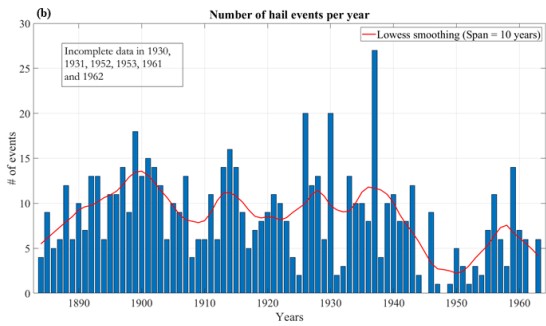

Figure 10: Panel (a): time series (blue line) of winter atmospheric pressure from 1884 to 1963. Each value (blue dot) is the average of the sub-daily observations measured during January, February and March. The red curve is the lowess smoothing filter, computed using a 10 years span. Panel (b): time series (blue vertical bar) of yearly hail events occurred at MVOBS from 1884 to 1963. The hail occurrence has been computed for every year of the investigated period, using the sub-daily observations of precipitation type. The red curve is the lowess smoothing filter, computed using a 10 years span. It should be noticed that such data were subject to a quality control procedure that did not include the homogenization.
