# Peer review of "Rescue and quality control of sub-daily meteorological data collected at Montevergine Observatory (Southern Apennines), 1884-1963"

_Earth System Science Data, 2020_

## Referee Comment (RC1) · Maria Carmen Beltrano (Referee) · 25 Mar 2020

The article is well structured. Methods and materials are well characterized and clearly describe the data set. Formal metadata are appropriate. The quality control procedures adopted are well described and ensure the high data quality. On the whole, the article is good, original and useful, appropriate to supporting the publication of the related data set. Considering the location of the meteorological observatory, the data set can be assessed very helpful in studying the climate of the central Mediterranean mountain area. Data set is correctly accessible via the given identifier, at

https://data.nodc.noaa.gov/cgi-bin/iso?id=gov.noaa.nodc:0205785#.

Some comments and suggestions of minor revisions or technical corrections: )( insert () delete

Page 3: row 73 and next ones: please, indicate here the four stations names (move from page 16 rows 499-500). However they are not the only ones: the auditor is aware of the existence of at least two other stations in Southern Italy which have functioned as observatories (three multiparametric observations per day), located in Foggia (Nigri Observatory, from the end of the 19th century to the 1970 and more) , Taranto (Ferrajolo Observatory, still functioning).

Page 4 row 103 data can (shade) light…. )shed( light row 114 drawn (is) section …. drawn )in( section row 123-124 e 126 please, titles translated in English enclosed in brackets

Page 5 row 138 The (tower measured) …. The )square-based tower measures( …. row 143 in the observatory (situation) room row 154 (suppressed) )interrupted(

Page 6 row 186: formatted )in tables( according to …. row 187-188 Please, review description: e.i. : Each table is related to a month; it is composed of two pages, the first column of each one lists the days, the first row the name of the parameters and observation time. )Each box( (On) of each column contains the value of…

Page 7 Row 204-207 please, add a figure of the more recent model of register. Row 220 please, add the reference for relationship formula.

Page 8 Row 247-256 please, add reference about snow/water conversion criteria .

Page 9 Row 288 please indicate which are remaining parameters

Page 11 Row 335 please, indicate that cloud cover parameter underwent only to gross error control and explain why.

Page 14 Row 443 reference: Brunet a or b?

Page 17 Row 538 (precepts) )rules(

Table 3: please indicate, below each QC code, the description: bad value, suspicious data, tolerance test, temporal incoherence, good value.

Figure 6: there aren't blue dots (outliers, QC=1)
* * *

---

## Short Comment (SC1) · 1 Apr 2020

Review to the paper ESSD-2020-38: Rescue and quality control of sub-daily meteorological data collected at Montevergine Observatory (Southern Apennines), 1884-1963

General impression: For climate analysis (past, present and future) it is essential to rescue past instrumental data. These data rescue projects involve a great effort and a very rigorous work. The paper and dataset presented contribute to recovery a significant amount of sub-daily data and especially data from the 19th century. They have

done a great job of digitizing data and recovering some metadata. The paper has an impact on the field. It has a high significance in this scientific field (climatological data rescue) and is within journal scope. I would recommend acceptance of this paper after a minor revision.

Review The writing is clear, concise and it is good English. Abstract: Brief and indicate the purpose of the work and what was done. Introduction: The purpose is clear. Goals and lacking in science are well illustrated. Section 2. Materials, data and methods: In an easier way it allows the reader to figure out the characteristics of the dataset and the methodology followed. Some comments: • Subsection 1: as important as data rescue is metadata rescue, metadata recovered is clear and give an image of the characteristics of observations through the time. It could be great if the authors can add some historical image. • Subsection 2: well explained and correct methodology. Only one comment, about ELP (lines 247-255) is required a reference about methodology applied and if it's possible add an schematic. Section 3: Quality control of digitized data • Lines 278-283: improve the way to flag the QC results, specially considering that not all variables pass all the tests. It is necessary to have a clear identificatory to flag (correct, suspicious and wrong values).

Section 4. Application examples of MVOBS sub-daily dataset: This section tries to give value to the dataset rescued, but I'm not in favour to keep this type of sections. On one hand, I think that for a specialist on this topic is obvious the value of the work done and on the other hand, (especially for the second part) they are trying to do a "climatological analysis" with potential inhomogeneous data (metadata reveals different potential breakpoints). So: • Maybe I'm in agree to keep the first part but, clearly stating that data used is not subjected to any homogenisation procedure and metadata indicates potential breakpoints. • I consider that is better to delete the second part. Section 5. Data availability: Really, good to know the availability of the data on an open access repository. Section 6. Conclusion. The conclusion is clearly stated and nice to read future aims. Tables: Clear to understand and information well summarized. Some

comments: • Table 3: Give information about what means QC =1 , QC = 2. . . Due to the way to name the flags, for cloud cover, rainfall and snowfall is not clear if 100 / 99.9 % of values are suspicious. . .

Figures: The figures are clear to understand and figure out the characteristics of the data rescued. Comments: • Figure 1. About the map, please add a reference on the repositories consulted. • Figure 4: The diagram is fine a clear, but the way to flag the results need to be improved (see comments above). • Figure 6: review the dots. According the graph and the text there are not blue dots • Figure 7: higher percentages needs to be well explained and maybe some visual information to explain this. • Figure 8: add a comment about data was only submitted to a QC not to a homogenisation procedure • Figure 9: I'm not sure that variability is only due to natural evolution. Needs to consider deleting this part. References Relevant and appropriate

Alba Gilabert Gallart PhD Centre for Climate Change (URV)

Please also note the supplement to this comment:
https://www.earth-syst-sci-data-discuss.net/essd-2020-38/essd-2020-38-SC1-supplement.pdf

---

## Editor Comment (EC1) · Kirsten Elger (Editor) · 10 Apr 2020

Dies Vincenzo Capozzi and co-authors,

We have already received two referee comments to your manuscript and I wanted to encourage you to take the opportunity of our public peer review and address the referee's comments in the public discussion while this is still ongoing (until May 1st)! Some authors of previous submissions even proceeded with the revision of the paper (within the public deiscussion) in cases when the referee reports were being submitted

at such an early stage. All referees are receiving notes whenever you are answering their questions and comments and would have the opportunity to react to them or taking a look at your revised manuscript.

Many thanks in advance and especially for having chosen ESSD kind regards,

Kirsten Elger

---

## Author Comment (AC1) · 15 Apr 2020

Dear Ms. Elger,

thank you for considering our manuscript for publication on ESSD and for your kind comment.

We are very pleased to follow your suggestion: it is our intention to submit a point-by-point reply to the referees comments, as well as a revised version of the paper, no later than 25 april, i.e. before the end of the public discussion.

[Figure]

We are also very grateful to both referees for their valuable suggestions and comments, which will help to improve our work.

Kind regards,

Vincenzo Capozzi On behalf of all Co-Authors of essd-2020-38 manuscript

––––––––––––––––––––––––––

---

## Author Comment (AC2) · 25 Apr 2020

Revision of

**"Rescue and quality control of sub-daily meteorological data collected at Montevergine Observatory (Southern Apennines), 1884–1963"**
**V. Capozzi, Y. Cotroneo, P. Castagno, C. De Vivo, G. Budillon**

**RC (X) =** Referee comment (number X)
**AR (X) =** Authors' reply (number X)
* * *
**Referee #1 (Maria Carmen Beltrano)**

**RC:** The article is well structured. Methods and materials are well characterized and clearly describe the data set. Formal metadata are appropriate. The quality control procedures adopted are well described and ensure the high data quality. On the whole, the article is good, original and useful, appropriate to supporting the publication of the related data set. Considering the location of the meteorological observatory, the data set can be assessed very helpful in studying the climate of the central Mediterranean mountain area. Data set is correctly accessible via the given identifier, at. https://data.nodc.noaa.gov/cgi-bin/iso?id=gov.noaa.nodc:0205785#

**AR:** Dear Ms. Beltrano, we are very grateful for your positive comments and relevant suggestions, which helped us improving our manuscript. The replies to your remarks are set out below. Moreover, we have modified the paper according to your recommendations.

**RC (1):** Page 3: row 73 and next ones: please, indicate here the four stations names (move from page 16 rows 499-500). However they are not the only ones: the auditor is aware of the existence of at least two other stations in Southern Italy which have functioned as observatories (three multiparametric observations per day), located in Foggia (Nigri Observatory, from the end of the 19th century to the 1970 and more), Taranto (Ferrajolo Observatory, still functioning).

**AR (1):** We have listed in the introduction the historical southern-Italy stations providing digitized data extending back prior 1960s, which can be found on the ISPDv4 database (rda.ucar.edu/datasets/ds132.2/index.html?sstn=17606&spart=exact#stationViewer).
The main text has been modified as follows:
"In accordance with the ISPDv4 database, there are only five other historical weather stations in southern Italy extending back several decades prior 1960s that had performed sub-daily multi-parametric observations and that may supply digitized data: Naples Capodimonte (40.88°N, 14.25°E), Foggia Nigri (41.46°N, 15.54°E), Taranto Ferrajolo, (40.47°N, 17.23°E), Palermo (38.10°N, 13.35°E) and Cagliari (39.20°N, 9.15°E). They are all located in coastal or near-coastal areas and only provide atmospheric pressure data with a temporal resolution of one observation per day (rda.ucar.edu/datasets/ds132.2/index.html?sstn=17606&spart=exact#stationViewer, last access: 29 January 2020). The digitized records available for these stations cover the period 1895-1940, except for the Taranto observatory whose time series spans a very limited time interval (1931-1939): for this reason, it has not been included in Fig. 1a.
In light of the above, the sub-daily data rescue activities carried out until now in southern Italy are incomplete. Furthermore, these datasets available in a digital format are only a small part of the larger amount of meteorological information stored in the original paper archives, both in terms of data temporal resolution and number of measured atmospheric parameters."
Regarding the two old observatories cited by the reviewer, the first one (Foggia Nigri) was already mentioned in the previous manuscript version (page 16, line 500); now we have also mentioned

Taranto Ferrajolo. However, we have decided not to include it in Fig. 1a due to the short record of available data (1931-1939), as shown in the ISPDv4 database.

**RC (2):** Page 4 row 103 data can (shade) light
row 114 drawn (is) section
row 123-124 e 126 please, titles translated in English enclosed in brackets

**AR (2):** We have modified the text according to the referee suggestions.

**RC (3):** Page 5 The (tower measured) . .. The )square-based tower measures( . .
row 143 in the observatory (situation) room
row 154 (suppressed) )interrupted(

**AR (3):** We have modified the text according to the referee recommendations.

**RC (4):** Page 6 formatted )in tables( according to . . ..
 row 187-188 Please, review description: e.i. : Each table is related to a month; it is composed of two pages, the first column of each one lists the days, the first row the name of the parameters and observation time. ). Each box( (On) of each column contains the value of. . .

**AR (4):** We have modified the description of handwritten meteorological registers stored in Montevergine Observatory.

**RC (5):** Page 7 Row 204-207 please, add a figure of the more recent model of register.
Row 220 please, add the reference for relationship formula.

**AR (5):** In Fig. 3, we have added two panels (c) and (d) to show a more recent register reporting the meteorological measurements collected in the second decade of January 1946 (see page 6 of this document). By comparing the added panels with a and b of Fig. 3, it can be observed that there are some differences with older register models. From 1944, a slightly different format was adopted in accordance with the new standards suggested by the Italian Central Office in Rome. A description of the January 1946 register structure has been added in the main text and is provided below for referee convenience:
"The meteorological registers of 1944-1961 period contain additional columns dedicated to other (sporadically measured) variables, such as snow depth, visibility and low-level clouds base height and quantity. Those registers have a different structure from the standard format described previously. Indeed, a single register consists of 72 pages (i.e. two pages for every decade of each month) and each page contains two tables. Panels (c) and (d) in Fig. 3 show the register structure for the decade of January 1946. In particular, the upper table on the left page (Fig. 3c) includes three-daily observations of atmospheric pressure, wind direction and force and cloud direction performed from day 11 to 20. The bottom table shows daily maximum and minimum temperature, three-daily observations of dry and wet bulb temperature, vapour pressure, relative humidity and finally the sum and average of thermometric measurements. The upper table on the right page (Fig. 3d), instead, contains sub-daily records of the sky conditions (cloud cover and type), accumulated rainfall, snow

depth and accumulated snowfall. In addition, daily summaries related to cloud cover, accumulated rainfall and snowfall, maximum 1-hour rainfall amount and precipitation duration (hours and minutes), are reported. The bottom table is dedicated to special notes concerning observed hydrometeors and meteorological phenomena."

Moreover, we have added the reference (Brombacher et al., 1960) for the relationship formula.

**RC (6):** Page 8 Row 247-256 please, add reference about snow/water conversion criteria.

**AR (6):** In the revised version of the manuscript, we added three references for the snow to equivalent liquid water conversion (Winiger, 2005; Egli, 2008; Egli et al., 2009).

**RC (7):** Page 9 Row 288 please indicate which are remaining parameters.

**AR (7):** In the revised version of our manuscript, we have indicated the meteorological parameters that only underwent through a basic manual inspection within the quality control procedure. These parameters are listed below: clouds direction, wind direction, wind speed, cloud type, visibility, low-level cloud base height and quantity, snow depth, precipitation duration and precipitation type.

**RC (8):** Page 11 Row 335 please, indicate that cloud cover parameter underwent only to gross error control and explain why.

**AR (8):** We have added the following sentence: "It should be noted that cloud cover data did not undergo tolerance and temporal coherence tests. The cloud amount was estimated by visual observations using a fixed reference scale. Due to the specific nature of this parameter and to its strong hour-to-hour variability, it is not possible to define climatological limits for outlier and anomalous jumps detection. Therefore, quality control for cloud cover includes only manual inspection and gross error test and it aims to assess the data plausibility and their consistency with other related meteorological parameters, such as cloud type and, when available, low-level clouds base height and quantity".

**RC (9):** Page 14 Row 443 reference: Brunet a or b?

**AR (9):** We apologize for the mistake. The correct reference is Brunet et al., 2014b.

**RC (10):** Page 17 Row 538 (precepts)

**AR (10):** We have modified the text according to referee's comment.

**RC (11):** Table 3: please indicate, below each QC code, the description: bad value, suspicious data, tolerance test, temporal incoherence, good value.

**AR (11):** In the first row of Table 3, and more clearly in the text, we have included a description for each QC flag (see page 5 of this document):

- bad data (QC = 8);
- suspicious data (QC = 9);
- good data, lower quality level (QC = 1), i.e. data that passed only gross error test
- good data, medium quality level (QC = 2), i.e. data that passed gross error and tolerance tests;
- good data, higher quality level (QC = 3), i.e. data that passed all statistical tests.

**RC (12):** Figure 6: there aren't blue dots (outliers, QC=1).

**AR (12):** In Figure 6, we show an application of QC procedure to relative humidity sub-daily observations collected in March 1901, where no data are flagged as QC = 1 (i.e. there aren't outliers). Therefore, we have deleted the phrase "blue dots (outliers, QC=1)" from the caption.

**List of new cited references**

Brombacher W. G., Johnson D. P., and Cross J. L.: Mercury Barometers and Manometers, NBS Mono.8, U.S. Govt. Printing Office, Washington, 1960.

Egli, L.: Spatial variability of new snow amounts derived from a dense network of Alpine automatic stations, Annals of Glaciology, 49, 51-55, https://doi.org/10.3189/172756408787814843, 2008.

Egli, L., Jonas, T., and Meister, R.: Comparison of different automatic methods for estimating snow water equivalent, Cold Regions Science and Technology, 57, Issues 2–3, 107-115, https://doi.org/10.1016/j.coldregions.2009.02.008, 2009.

The International Surface Pressure Databank version 4, Interactive Station Viewer: https://rda.ucar.edu/datasets/ds132.2/index.html?sstn=17606&spart=exact#stationViewer, last access: 29 January 2020.

Winiger, M., Gumpert, M., and Yamout, H.: Karakorum–Hindukush–western Himalaya: assessing high-altitude water resources, Hydrol. Process. 19, 2329–2338, https://doi.org/10.1002/hyp.5887, 2005.

**List of modified tables according to referee suggestion. We highlighted our changes in yellow.**

**Table 3.** Results of quality control tests applied to MVOBS sub-daily meteorological data. Each column show the percentage of data flagged as QC = 8, QC =9, QC = 1, QC = 2 and QC = 3. It should be noted that cloud cover data underwent only manual inspection and gross error test, whereas rainfall and snowfall measurements quality was evaluated according to manual inspection, gross error and tolerance tests.

| Parameter | % of QC = 8 bad data | % of QC = 9 suspicious data | % of QC = 1 good data (lower quality level) | % of QC = 2 good data (medium quality level) | % of QC = 3 good data (higher quality level) |
|---|---|---|---|---|---|
| Dry bulb temperature | 0.0 | 13.0 | 0.2 | 0.6 | 86.2 |
| Wet bulb temperature | 0.0 | 13.1 | 0.4 | 0.8 | 85.7 |
| Atmospheric pressure | 0.0 | 0.0 | 0.4 | 1.3 | 98.3 |
| Vapour pressure | 0.4 | 12.8 | 0.4 | 1.1 | 85.3 |
| Relative humidity | 0.3 | 12.8 | 0.4 | 1.6 | 84.8 |
| Cloud cover | 0.0 | 0.0 | 100.0 | Not applied | Not applied |
| Rainfall | 0.0 | 0.0 | 0.1 | 99.9 | Not applied |
| Snowfall | 0.0 | 0.0 | 0.1 | 99.9 | Not applied |

**List of modified figures according to referee suggestions. We highlighted our changes in yellow.**

[Figure]

Figure 3: Upper (a, b) and middle (c, d) panels show an example of original data source (March 1892 and January 1946, respectively). Each row accounts for the observations of a specific day, including their average on decadal and monthly basis, whereas each column is devoted to the records of a determined parameter at a specific hour of the day. The bottom panel (e) is an example (referred to data collected in March 1892) of the template used in the data digitisation. The rows are designed to match the location of the data in the original source.

---

## Author Comment (AC4) · 25 Apr 2020

Dear Ms. Gilabert Gallart,

thank you for the valuable comments provided about our manuscript. The replies to the open questions can be accessed through the following link:

https://editor.copernicus.org/index.php/essd-2020-38-AC3.pdf?_mdl=msover_md&_jrl=386&_lcm=oc108lcm109w&_acm=

Best regards,

[Figure]

Vincenzo Capozzi

On behalf of all Co-Authors

---

## Author Response (AR1)

Revision of

**"Rescue and quality control of sub-daily meteorological data collected at Montevergine Observatory (Southern Apennines), 1884–1963"**
**V. Capozzi, Y. Cotroneo, P. Castagno, C. De Vivo, G. Budillon**

**RC (X)** = Referee comment (number X)
**AR (X)** = Authors' reply (number X)

**Referee #1 (Maria Carmen Beltrano)**

**RC:** The article is well structured. Methods and materials are well characterized and clearly describe the data set. Formal metadata are appropriate. The quality control procedures adopted are well described and ensure the high data quality. On the whole, the article is good, original and useful, appropriate to supporting the publication of the related data set. Considering the location of the meteorological observatory, the data set can be assessed very helpful in studying the climate of the central Mediterranean mountain area. Data set is correctly accessible via the given identifier, at. https://data.nodc.noaa.gov/cgi-bin/iso?id=gov.noaa.nodc:0205785#

**AR:** Dear Ms. Beltrano, we are very grateful for your positive comments and relevant suggestions, which helped us improving our manuscript. The replies to your remarks are set out below. Moreover, we have modified the paper according to your recommendations and we highlighted our changes in the main text in yellow.

**RC (1):** Page 3: row 73 and next ones: please, indicate here the four stations names (move from page 16 rows 499-500). However they are not the only ones: the auditor is aware of the existence of at least two other stations in Southern Italy which have functioned as observatories (three multiparametric observations per day), located in Foggia (Nigri Observatory, from the end of the 19th century to the 1970 and more), Taranto (Ferrajolo Observatory, still functioning).

**AR (1):** We have listed in the introduction (page 3, lines 73-75 of the new version of the manuscript) the historical southern-Italy stations providing digitized data extending back prior 1960s which can be found on the ISPDv4 database
(rda.ucar.edu/datasets/ds132.2/index.html?sstn=17606&spart=exact#stationViewer).
Regarding the two old observatories cited by the reviewer, the first one (Foggia Nigri) was already mentioned in the previous manuscript version (page 16, line 500); now we have also mentioned Taranto Ferrajolo. However, we have decided not to include it in Fig. 1a (page 3, lines 78-79) due to the short record of available data (1931-1939), as shown in the ISPDv4 database.

**RC (2):** Page 4 row 103 data can (shade) light
row 114 drawn (is) section
row 123-124 e 126 please, titles translated in English enclosed in brackets

**AR (2):** We have modified the text according to the referee suggestions (see page 4, lines 111 and 123 and page 5, lines 132-133 and 135).

**RC (3):** Page 5 The (tower measured) . .. The )square-based tower measures( . .
row 143 in the observatory (situation) room
row 154 (suppressed) )interrupted(

**AR (3):** We have modified the text according to the referee recommendations (see page 5, lines 147 and 151; page 6, line 164).

**RC (4):** Page 6 formatted )in tables( according to . . ..
 row 187-188 Please, review description: e.i. : Each table is related to a month; it is composed of two pages, the first column of each one lists the days, the first row the name of the parameters and observation time. ). Each box( (On) of each column contains the value of. . .

**AR (4):** We have modified the description of handwritten meteorological registers stored in Montevergine Observatory (page 7, lines 196-198).

**RC (5):** Page 7 Row 204-207 please, add a figure of the more recent model of register.
Row 220 please, add the reference for relationship formula.

**AR (5):** In Fig. 3, we have added two panels (c) and (d) to show a more recent register reporting the meteorological measurements collected in the second decade of January 1946. By comparing the added panels with a and b of Fig. 3, it can be observed that there are some differences with older register models. From 1944, a slightly different format was adopted in accordance with the new standards suggested by the Italian Central Office in Rome. A description of the January 1946 register structure has now been provided at page 7 (lines 222-232).
Moreover, we have added the reference (Brombacher et al., 1960) for the relationship formula (see page 8, line 243).

**RC (6):** Page 8 Row 247-256 please, add reference about snow/water conversion criteria.

**AR (6):** In the revised version of the manuscript (page 9, line 275), we added three references for the snow to equivalent liquid water conversion (Winiger, 2005; Egli, 2008; Egli et al., 2009).

**RC (7):** Page 9 Row 288 please indicate which are remaining parameters.

**AR (7):** We have indicated the meteorological parameters that only underwent through a basic manual inspection within the quality control procedure (page 10, lines 317-318).

**RC (8):** Page 11 Row 335 please, indicate that cloud cover parameter underwent only to gross error control and explain why.

**AR (8):** We have added the following sentence at page 12 (Lines 372-377): "It should be noted that cloud cover data did not undergo tolerance and temporal coherence tests. The cloud amount was estimated by visual observations using a fixed reference scale. Due to the specific nature of this parameter and to its strong hour-to-hour variability, it is not possible to define climatological limits for outlier and anomalous jumps detection. Therefore, quality control for cloud cover includes only manual inspection and gross error test and it aims to assess the data plausibility and their consistency with other related meteorological parameters, such as cloud type and, when available, low-level clouds base height and quantity"

**RC (9):** Page 14 Row 443 reference: Brunet a or b?

**AR (9):** We apologize for the mistake. The correct reference is Brunet et al., 2014b (page 15, line 488).

**RC (10):** Page 17 Row 538 (precepts)

**AR (10):** We have modified the text according to referee's comment (see page 18, line 585).

**RC (11):** Table 3: please indicate, below each QC code, the description: bad value, suspicious data, tolerance test, temporal incoherence, good value.

**AR (11):** In the first row of Table 3, and more clearly in the text (page 10, Lines 307-311), we have included a description for each QC flag:
- bad data (QC = 8);
- suspicious data (QC = 9);
- good data, lower quality level (QC = 1), i.e. data that passed only gross error test
- good data, medium quality level (QC = 2), i.e. data that passed gross error and tolerance tests;
- good data, higher quality level (QC = 3), i.e. data that passed all statistical tests.

**RC (12):** Figure 6: there aren't blue dots (outliers, QC=1).

**AR (12):** In Figure 6 (now labelled Fig. 7 in the new version of the manuscript) we show an application of QC procedure to relative humidity sub-daily observations collected in March 1901, where no data are flagged as QC = 1 (i.e. there aren't outliers). Therefore, we have deleted the phrase "blue dots (outliers, QC=1)" from the caption.

**List of new cited references**

Brombacher W. G., Johnson D. P., and Cross J. L.: Mercury Barometers and Manometers, NBS Mono.8, U.S. Govt. Printing Office, Washington, 1960.

Egli, L.: Spatial variability of new snow amounts derived from a dense network of Alpine automatic stations, Annals of Glaciology, 49, 51-55, https://doi.org/10.3189/172756408787814843, 2008.

Egli, L., Jonas, T., and Meister, R.: Comparison of different automatic methods for estimating snow water equivalent, Cold Regions Science and Technology, 57, Issues 2–3, 107-115, https://doi.org/10.1016/j.coldregions.2009.02.008, 2009.

The International Surface Pressure Databank version 4, Interactive Station Viewer: https://rda.ucar.edu/datasets/ds132.2/index.html?sstn=17606&spart=exact#stationViewer, last access: 29 January 2020.

Winiger, M., Gumpert, M., and Yamout, H.: Karakorum–Hindukush–western Himalaya: assessing high-altitude water resources, Hydrol. Process. 19, 2329–2338, https://doi.org/10.1002/hyp.5887, 2005.

**RC:** General impression: For climate analysis (past, present and future) it is essential to rescue past instrumental data. These data rescue projects involve a great effort and a very rigorous work. The paper and dataset presented contribute to recovery a significant amount of sub-daily data and especially data from the 19th century. They have done a great job of digitizing data and recovering some metadata. The paper has an impact on the field. It has a high significance in this scientific field (climatological data rescue) and is within journal scope. I would recommend acceptance of this paper after a minor revision.

**AR:** Dear Ms. Gilabert Gallart, we are grateful for your positive evaluation of our study. We are glad to clarify the open questions and modify the paper accordingly to your recommendations. We highlighted our changes on the main text in yellow.
RC (X) stands for Referee comment (number X).
AR (X) stands for Authors' reply (number X).

**RC (1):** Section 2. Materials, data and methods: In an easier way it allows the reader to figure out the characteristics of the dataset and the methodology followed.
Some comments: -Subsection 1: as important as data rescue is metadata rescue, metadata recovered is clear and give an image of the characteristics of observations through the time. It could be great if the authors can add some historical image.

**AR (1):** We have added some historical pictures of Montevergine Observatory in the Appendix A (page 20, Figure A1). We have also included a recent panoramic image of Montevergine Abbey, to highlight that Montevergine Observatory is surrounded by a natural high-altitude environment, whose features have remained unchanged over time.

**RC (2) 190:** Subsection 2: well explained and correct methodology. Only one comment, about ELP (lines 247-255) is required a reference about methodology applied and if it's possible add a schematic.

**AR (2):** In the revised version of the manuscript (page 9, line 275), we have added three references about snow to equivalent liquid water conversion (Winiger, 2005; Egli, 2008; Egli et al., 2009). Moreover, to better explain the strategy adopted to determine the equivalent liquid precipitation (ELP) parameter, we have produced a new figure (Fig. 4 in the revised version of the manuscript). This figure shows an adapted extract of the rescued MVOBS dataset (available on NOAA's NCEI repository, Capozzi et al., 2019) focusing on the precipitation measurements collected between 28 and 30 January 1956. In addition, the different scenarios involving an estimation of ELP discussed in the main text (page 9, lines 270-283) are illustrated with specific reference to the sub-daily data recorded in this time segment.

**RC (3)** Quality control of digitized data - Lines 278-283: improve the way to flag the QC results, specially considering that not all variables pass all the tests. It is necessary to have a clear identificatory to flag (correct, suspicious and wrong values).

**AR (3):** We are grateful to the referee for this comment, which allow us to better clarify the meaning of the different QC flags considered in our study. The QC labels and the related description are listed below:

- bad data (QC = 8), i.e. data that did not satisfy the gross error test;
- suspicious data (QC = 9), i.e. data that did not passed the manual inspection or that did not satisfy the inter-variable check;

- good data, lower quality level (QC = 1), i.e. data that passed only gross error test
- good data, medium quality level (QC = 2), i.e. data that passed gross error and tolerance tests;
- good data, higher quality level (QC = 3), i.e. data that passed all statistical tests.

According to this classification of QC flags, data that have passed at least one objective statistical check are defined as "good" and are associated to a quality level (ranging from low to high) that is a function of the number of statistical tests satisfied. We have better stressed this concept in the main text (see page 10, lines 307-311). Moreover, we have improved figure 5, which offers a schematic diagram of the quality-control procedure, and table 3 that summarizes the results of quality control tests. In table 3 caption, we have specified that cloud cover parameter did not underwent tolerance and temporal coherence tests. As explained at page 12 (lines 372-377), cloud amount was estimated by visual observations using a fixed reference scale. Due to the specific nature of this parameter and its strong hour-to-hour variability, it is not possible to define climatological limits for outlier and anomalous jumps detection. Therefore, quality control for cloud cover includes only manual inspection and gross error test and it aims to assess the data plausibility and their consistency with other related meteorological parameters, such as cloud type and, when available, low-level clouds base height and quantity. In addition, it should be noted that, since rainfall and snowfall data are time-integrated values, they were not analyzed in terms of plausible rate of change and therefore for those parameters the highest quality flag is QC = 2.

**RC (4)** Section 4. Application examples of MVOBS sub-daily dataset: This section tries to give value to the dataset rescued, but I'm not in favour to keep this type of sections. On one hand, I think that for a specialist on this topic is obvious the value of the work done and on the other hand, (especially for the second part) they are trying to do a "climatological analysis" with potential inhomogeneous data (metadata reveals different potential breakpoints). So: -Maybe I'm in agree to keep the first part but, clearly stating that data used is not subjected to any homogenisation procedure and metadata indicates potential breakpoints. -I consider that is better to delete the second part.

**AR (4):** In our opinion, section 4.2 can be of interest to many readers because it provides concrete evidence on the possible applications of Montevergine sub-daily data in climatological studies. Therefore, we have decided to leave this section. The aim of section 4.2 is to emphasize the potential use of multi-parametric Montevergine time series to analyze "less-studied" atmospheric variables, whose past climate variability is largely unknown especially in Mediterranean region. The hail events frequency of occurrence, showed in Fig. 10 (of the updated version of the manuscript) is a relevant example: in many regions, such as Italy (Baldi, 2014), the scarcity of historical information does not allow to build a solid long-term climatology for this parameter. This can also be considered valid for the snowfall amount, which is a very important parameter for mountain environment, also from a hydrological perspective.
We acknowledge that a single isolated time-series (although with many distinguish features, such as Montevergine) can only give a partial contribution to the climate reconstruction. However, we hope that our effort may be an incentive for future initiatives aimed at rescuing historical sub-daily multi-parametric time series.
To conclude, as pointed out in the updated version of the manuscript (page 16, lines 499-503), this section has only "illustrative purposes": it is not our intention to perform a climatologic analysis, which requires homogenized data as rightly highlighted by referee. This aspect has been further clarified at page 17 (lines 546-547).

**RC (5)** Table 3: Give information about what means QC =1, QC = 2. . . Due to the way to name the flags, for cloud cover, rainfall and snowfall is not clear if 100 / 99.9 % of values are suspicious.

**AR (5)**: According to referee's suggestion, in the first row of Table 3 we have added a description for each QC flag:
•       bad data (QC = 8);
•       suspicious data (QC = 9);
•       good data, lower quality level (QC = 1), i.e. data that passed only gross error test
•       good data, medium quality level (QC = 2), i.e. data that passed gross error and tolerance tests;
•       good data, higher quality level (QC = 3), i.e. data that passed all statistical tests.

As explained in the reply to the RC (3), cloud cover, rainfall and snowfall data did not undergo the entire QC procedure. Cloud cover was checked only with manual inspection and gross error test (i.e, the maximum quality level is QC=1), whereas rainfall and snowfall data quality was evaluated through manual inspection, gross error and tolerance tests (i.e. the maximum quality level is QC=2). The results obtained for these parameters (100% of cloud cover data were flagged as QC=1 and 99.9% of rainfall and snowfall data were labelled as QC=2) should be interpreted as very encouraging signs about their reliability. When other sub-daily time series collected in Southern Italy will become available, an additional quality control assessment will be performed as a mean of spatial consistency check.

**RC (6)** Figure 1. About the map, please add a reference on the repositories consulted.

**AR (6)**: In the previous manuscript version, caption of Fig. 1 had two references about the consulted repositories (Ashcroft et al., 2018; Compo et al., 2019).
We have added a further reference to the interactive tool of The International Surface Pressure Databank version 4 (rda.ucar.edu/datasets/ds132.2/index.html?sstn=1
7606&spart=exact#stationViewer), that allows to search for available atmospheric pressure sub-daily data according to the period and to the geographical area of interest.

**RC (7)** Figure 4: The diagram is fine a clear, but the way to flag the results need to be improved (see comments above).

**AR (7):** We have modified the diagram of this figure (Fig. 5 in the new version of the manuscript) according to the referee suggestions in reply to RC (3).

**RC (8)** Figure 6: review the dots. According the graph and the text there are not blue dots.

**AR (8):** In Figure 6 (now labelled Fig. 7 in the new version of the manuscript) we show an application of QC procedure to relative humidity sub-daily observations collected in March 1901, where no data are flagged as QC = 1 (i.e. there aren't outliers). Therefore, we have deleted the phrase "blue dots (outliers, QC=1)" from the caption.

**RC (9)** Figure 7: higher percentages needs to be well explained and maybe some visual information to explain this.

**AR (9):** We have added a discussion about the results showed by figure 7, which is now numbered as Fig. 8 in the new manuscript version (see page 15, Lines 471-479). The higher percentages found in some of the sub-periods (1919-1923, 1924-1928, 1944-1948, and 1949-1953) are related to the very high number of observations flagged as QC=9 (i.e. suspicious data) after visual inspection. In

these time segments, data quality was affected by some impairments in thermo-psychrometric measurements, mainly caused by human errors. A brief discussion on such issues was supplied in section 3.1 (page 11, lines 340-342).

We have provided additional details by adding two panels to Fig. 8, labelled (b) and (c), where we show the frequency distribution of dry-bulb temperature measurements in 1919-1925 and 1948-1950 period respectively. In Figure 8b, it can be clearly seen that even temperature values have an absolute frequency much higher than odd ones. Whereas, histogram in Fig. 8c, which only shows temperature records between 10 and 20°C, highlights an anomalous high frequency in the integer temperature values recorded.

**RC (10)** Figure 8: add a comment about data was only submitted to a QC not to a homogenization procedure.

**AR (10)**: We modified the caption of this figure (now labelled Fig. 9 in the new version of the manuscript) by adding the following sentence (see pag. 36, Lines 865-866): "It should be noticed that such data were subject to a quality control procedure that did not include the homogenization".

**RC (11)** Figure 9: I'm not sure that variability is only due to natural evolution. Needs to consider deleting this part.

**AR (11):** We partially agree with the reviewer. We are aware that artefacts caused by the potential inhomogeneity revealed by metadata probably undermine the climatic signal presented in this figure (which is Fig. 10 in the revised paper). Therefore, at this stage, it is not possible to achieve conclusions about climatic variability and trends from Montevergine sub-daily data. This aspect has been highlighted at pages 16 (Lines 499-503) and 17 (Lines 546-547).

However, as stated in the reply to RC (4), we feel appropriate leaving this section in our manuscript, because it shows, from a qualitative perspective, some possible future applications of Montevergine sub-daily records in climate fields, with a particular emphasis on some meteorological parameters whose historical variability is largely unknown.

**List of cited references**

Ashcroft, L., Coll, J.R., Gilabert, A., Domonkos, P., Brunet, M., Aguilar, E., Castella, M., Sigro, J., Harris, I., Unden, P., and Jones, P.: A rescued dataset of sub-daily meteorological observations for Europe and the southern Mediterranean region, 1877-2012, Earth Syst. Sci. Data, 10, 1613-1635, https://doi.org/10.5194/essd-10-1613-2018, 2018.

Baldi, M., Ciardini, V., Dalu, J.D., Filippis, T.D., Maracchi, G., and Dalu, G.: Hail occurrence in Italy: towards a national database and climatology, Atmos. Res., 138, 268–277, https://doi.org/10.1016/j.atmosres.2013.11.012, 2014.

Capozzi, V., Cotroneo, Y., Castagno, P., De Vivo, C., Komar, A., Guariglia, R., Budillon,: Sub-daily meteorological data collected at Montevergine Observatory (Southern Apennines), Italy from 1884-01-01 to 1963-12-31 (NCEI Accession 0205785). NOAA National Centers for Environmental Information. Dataset. https://doi.org/10.25921/cx3g-rj98, 2019.

Compo, G. P., Whitaker, J. S., Sardeshmukh, P. D., Matsui, N., Allan, R. J., Yin, X., Gleason, B. E., Vose, R. S., Rutledge, G., Bessemoulin, P., Brönnimann, S., Brunet, M., Crouthamel, R. I., Grant, A. N., Groisman, P. Y., Jones, P. D., Kruk, M. C., Kruger, A. C., Marshall, G. J., Maugeri, M., Mok, H. Y., Nordli, Ø., Ross, T. F., Trigo, R. M., Wang, X. L., Woodruff, S. D., and Worley, S. J.: The Twentieth Century Reanalysis

Egli, L.: Spatial variability of new snow amounts derived from a dense network of Alpine automatic stations, Annals of Glaciology, 49, 51-55, https://doi.org/10.3189/172756408787814843, 2008.

Egli, L., Jonas, T., and Meister, R.: Comparison of different automatic methods for estimating snow water equivalent, Cold Regions Science and Technology, 57, Issues 2–3, 107-115, https://doi.org/10.1016/j.coldregions.2009.02.008, 2009.

The International Surface Pressure Databank version 4, Interactive Station Viewer: https://rda.ucar.edu/datasets/ds132.2/index.html?sstn=17606&spart=exact#stationViewer, last access: 29 January 2020.

Winiger, M., Gumpert, M., and Yamout, H.: Karakorum–Hindukush–western Himalaya: assessing high-altitude water resources, Hydrol. Process. 19, 2329–2338, https://doi.org/10.1002/hyp.5887, 2005.

---

## Author Response (AR2)

Revision of

**"Rescue and quality control of sub-daily meteorological data collected at Montevergine Observatory (Southern Apennines), 1884–1963"**
*V. Capozzi, Y. Cotroneo, P. Castagno, C. De Vivo, G. Budillon*

**EC (X)** = Editor comment (number X)
**AR (X)** = Authors' reply (number X)

**EC (1):** Dear Vicenzo and co-authors, many thanks for the very good revision of the manuscript. I am happy to accept the paper for publication after addressing two minor change requests related to RC2 of Referee #1 (i.e. the requested translation of the original pamphlets referred to on page 5, lines 132-133 and 135):
I have seen that you translated the original names in English in the revised version as requested by the referee. My understanding of RC 2, however, was to provide the English translation in addition to the original name. I also think that keeping the Italian name and adding the English translation in brackets would be much better to allow people finding the original source.
I also suggest a different translation of the first reference from "In the fiftieth of Montevergine meteorological observatory" to "The fiftieth anniversary of Montevergine meteorological observatory"
I suggest to change the sentences to:

p. 5, line 132/133: "....and a pamphlet entitled "Nel Cinquantenario dell'Osservatorio Meteorologico di Montevergine" (The fiftieth anniversary of Montevergine meteorological observatory), published in 1934 on ...."

p. 5, line 135: ".... from another pamphlet named "L'Osservatorio meteorologico di Montevergine" (Montevergine meteorological observatory) that was published in 1983..."

many thanks, also for choosing ESSD
and best regards,

Kirsten Elger

**AR (1):** Dear Ms. Elger, we are very grateful for your positive evaluation of our revision work. We apologize for the misunderstanding about the RC(2) of referee #1. We have modified the two sentences according to your suggestions. Note that the right title of the second pamphlet is "Osservatorio Meteorologico Santuario di Montevergine" (Montevergine Abbey meteorological observatory); it was published in 1984. We apologize for the typos. Our changes in the main text are highlighted in yellow.

Best regards,

Vincenzo Capozzi
On the Behalf of all co-authors